

# Impacts of Tibetan Plateau uplift on atmospheric dynamics and associated precipitation δ$^{18}$O

**S. Botsyun[1], P. Sepulchre[1], C. Risi[2], and Y. Donnadieu[1]**

[1]{Laboratoire des Sciences du Climat et de l'Environnement, LSCE/IPSL, CEA-CNRS-UVSQ, Université Paris-Saclay, Gif-sur-Yvette, France}

[2]{Laboratoire de Météorologie Dynamique, LMD/IPSL, UPMC, CNRS, Paris, France}

Correspondence to: S. Botsyun (svetlana.botsyun@lsce.ipsl.fr)

## Abstract

Paleoelevation reconstructions of mountain belts have become a focus of modern science since surface elevation provides crucial information for understanding both geodynamic mechanisms of Earth's interior and influence of mountains growth on climate. Stable oxygen isotopes paleoaltimetry is one of the most popular techniques nowadays, and relies on the difference between δ$^{18}$O of paleo-precipitation reconstructed using the natural archives, and modern measured values for the point of interest. Our goal is to understand where and how complex climatic changes linked with the growth of mountains affect δ$^{18}$O in precipitation. For this purpose, we develop a theoretical expression for the precipitation composition and we use the isotope-equipped atmospheric general circulation model LMDZ-iso. Experiments with reduced height over the Tibetan Plateau and the Himalayas have been designed. Our results show that the isotopic composition of precipitation is very sensitive to climate changes related with the growth of the Himalayas and Tibetan Plateau, notably changes in relative humidity and precipitation amount. The relative contribution of controlling factors and their magnitude differ depending on the uplift stage and the region considered. Thus future paleoaltimetry studies should take into account constraints on climatic factors to avoid misestimating ancient altitudes.



**1    Introduction**
Despite ongoing debates regarding the thermal and mechanical nature of mechanisms
involved (Boos, 2015; Chen et al., 2014), the Himalayas and the Tibetan Plateau (hereafter
TP) have long been considered to exert major influences on Asian atmospheric dynamics,
notably by reinforcing South Asian monsoon and driving subsidence ultimately leading to
onsets of deserts over Central Asia (Rodwell and Hoskins, 2001; Broccoli and Manabe,
1992). Thus, reconstructing the history of Himalayas and TP uplift appears crucial to
understand long-term climate evolution of Asia. On the other hand, topography uplift of TP is
ultimately driven by collision between India and Asia continents (Molnar et al., 2010),
making the timing and scale of surface elevation growth widely used for reconstructing the
rate and style of this tectonic plates convergence (eg. Royden et al., 2008; Tapponnier et al.,

12  2001).

Elevation reconstructions for the Tibetan Plateau and Himalayas are based on fossil-leaf
morphologies (eg. Antal, 1993; Forest et al., 1999; Khan et al., 2014), pollen (Dupont-Nivet
et al., 2008) correlation between stomatal density and the decrease in $CO_2$ partial pressure
with altitude (McElwain, 2004) and carbonate oxygen isotopic compositions (Currie et al.,
2005; DeCelles et al., 2007; Garzione et al., 2000; Polissar et al., 2009; Rowley and Currie,
2006; Saylor et al., 2009; Xu et al., 2013; Zhuang et al., 2014; Li et al., 2015). In contrast to
paleobotanical methods, oxygen isotope paleoaltimetry has been widely applied for the
Cenozoic. Carbonate $\delta^{18}O$ is related to topography change using $\delta^{18}O$-elevation relationship.
These relationships have been calibrated both empirically (eg. Garzione et al., 2000; Poage
and Chamberlain, 2001) and theoretically, using basic thermodynamic principles, including
Rayleigh distillation, that govern isotopic fractionation processes (Rowley and Garzione,
2007; Rowley et al., 2001).
The difference between paleoprecipitation $\delta^{18}O$ detected from natural archives and modern
values of the site of interest is identified with the effect of the surface uplift in numerous
recent studies (Currie et al., 2005; Cyr et al., 2005; Ding et al., 2014; Hoke et al., 2014;
Mulch, 2016; Rowley and Currie, 2006; Rowley et al., 2001; Xu et al., 2013). In the absence
of direct measurements of "paleo" altitude-$\delta^{18}O$ relationship *in situ*, stable-isotope
paleoaltimetry is potentially hampered by the fact that the presumed constancy of altitude-
$\delta^{18}O$ relationships through time might not be valid. For instance for the Andes, not
considering the impact of uplift on climate dynamics and related $\delta^{18}O$ values has been shown



to produce errors in paleoelevation reconstruction reaching up to ± 50% (Ehlers and Poulsen,
2009; Poulsen et al., 2010). Regional climate variables and associated isotopic signal in
precipitation can also be affected by global climate change (Jeffery et al., 2012; Poulsen and
Jeffery, 2011). Moreover, it has been suggested that climate-driven changes in surface ocean
$\delta^{18}O$ through the Cenozoic can also influence recorded values of precipitation $\delta^{18}O$ over the
continent (Ding et al., 2014). Over TP, mismatches between paleoelevation estimations from
palynological and stable isotope data (eg. Sun et al., 2014) could be related to complex
climatic changes and associated variations of altitude-$\delta^{18}O$ relationship linked to the uplift,
still a detailed assessment of the consequences of topographic changes on precipitation $\delta^{18}O$ is
lacking.
Spatial distribution of isotopes in precipitation was described using various types of models,
from one-dimensional to three-dimensional general circulation (Craig, 1961; Dansgaard,
1964; Gedzelman and Arnold, 1994; Risi et al., 2010; Stowhas and Moyano, 1993). Such
modelling studies show how large-scale Asian monsoon circulation influence precipitation
$\delta^{18}O$ (He et al., 2015; LeGrande and Schmidt, 2009; Pausata et al., 2011; Vuille et al., 2005).
At the global scale, precipitation $\delta^{18}O$ has been shown to be affected by several factors other
than elevation, including mixing between air masses (Ehlers and Poulsen, 2009; Gat, 1996),
large-scale subsidence (e.g. Frankenberg et al., 2009), continental recycling (Lee et al., 2012;
Risi et al., 2013), deep convection (Risi et al., 2008), and enrichments linked to global
warming (Poulsen and Jeffery, 2011). Numerous studies have investigated the impact of
Asian topography on climate change, including the monsoon intensification (ex. Harris, 2006;
Kutzbach et al., 1989; Raymo and Ruddiman, 1992; Zhang et al., 2015; Zhisheng et al., 2015)
and Asian interior aridification onset (Broccoli and Manabe, 1992; Liu et al., 2015).
Nonetheless the linkage between these "climatic parameters" altered by the growth of TP and
their influence on the isotopic signal remain unclear. In this article we use numerical
modelling to provide some insights.
**2   Methods**
**2.1   Model simulations**
We use an Atmospheric General Circulation model (GCM) developed at Laboratoire de
Météorologie Dynamique, Paris, France with isotopes-tracking implement, called LMDZ-iso



(Risi et al., 2010). LMDZ-iso is derived from the LMDz model (Hourdin et al., 2006) that has
been used for numerous future and paleoclimate studies (Ladant et al., 2014; Pohl et al., 2014;
Sepulchre et al., 2006). Water in a condensed form and its vapour are advected by the Van
Leer advection scheme (Van Leer, 1977). Isotopic processes in LMDZ-iso are documented in
(Risi et al., 2010). Evaporation over land is assumed not to fractionate, given the simplicity of
the model surface parameterisation (Risi et al., 2010). Yao et al. (2013) have provided a
precise description of rainfall patterns over the TP, and showed LMDZ-iso ability to simulate
atmospheric dynamics and reproduce rainfall and $\delta^{18}O$ patterns consistent with data over this
region.
LMDZ-iso is also equipped with water tagging capabilities, allowing to quantify different
moisture contributions from continental and oceanic evaporation sources. The advantage of
this technique compared to typical back-trajectories methods is that it tracks the water rather
than air masses, thus taking into account effects of phase changes. In our simulations five
potential moisture sources are considered: (1) continental sources, (2) Indian Ocean, (3)
Atlantic Ocean, (4) Mediterranean Sea, and (5) Pacific Ocean.
We use a model configuration with 96 grid points in longitude, 72 in latitude and 19 vertical
layers, with the first four layers in the first kilometer above the surface. LMDZ-iso has a
stretchable grid that allows increased spatial resolution over a defined region. In our case, it
gives an averaged resolution of ~100 km over central Asia, which is a good trade-off between
a reasonable computing time and a spatial resolution that adequately represents main features
of TP topography.
Here we report results from three experiments designed to isolate the influence of Asian
topography on climate and isotopic composition of precipitation. Topography is derived from
a 10-minute US Navy dataset and interpolated to the model grid. The control run (MOD) is a
pre-industrial run, i.e. initialized with boundary conditions (insolation, greenhouse gases, sea
surface temperatures (SSTs), topography) kept at pre-industrial values. For the two other
experiments, we keep all boundary conditions (including albedo, rugosity, and vegetation
distribution) similar to those in MOD run, except for the topography. We reduce the altitude
over the area covering the Tibetan Plateau, Himalayas and a part of surrounding mountains:
Tian Shan, Pamir, Kunlun and Hindu Kush to 50% of modern elevations (intermediate, INT
case) and to 250-m elevation (low, LOW case) (Fig. 1). SSTs for all runs come from the
AMIP dataset (monthly SSTs averaged from 1979 to 1996; Taylor et al., 2000). Each





experiment has been run for 20 years. We analyse seasonal means over the last 18 years, as
the two first years are extracted for spin-up.

### 2.2 Theoretical framework for the precipitation composition

Our goal is to understand to what extent topography changes explain the precipitation $\delta^{18}$O
signal over TP (i.e. the direct topography effect) and what part of this signal depends on other
climate processes. To do so, we develop a theoretical expression for the precipitation
composition.
To the first order, the $\delta^{18}$O composition of the precipitation $R_p$ follows that of the vapour $R_v$.
Deviations from the vapour composition, $\varepsilon = R_p - R_v$, are associated with local condensational
or post condensational process.
$R_p = R_v + \varepsilon$ (1)
In an idealized framework of an isolated air parcel transported from an initial site at low
altitude to the site of interest (Fig. 2), the vapour composition can be predicted by Rayleigh
distillation:
$R_v = R_{vi} \cdot f^{(\alpha-1)}$ (2)
where $R_{vi}$ is the initial composition of the vapour at the initial site, $\alpha$ is the fractionation
coefficient, that depends on temperature and on the water phase (Majzoube, 1971; Merlivat
and Nief, 1967), and $f$ is the residual fraction of the vapour at the site of interest relatively to
the initial site. We take the initial site as characterised by a temperature and humidity $T_0$ and
$q_0$. Under these conditions, we note $R_{v0}$ the theoretic isotopic composition that it would have
if all the vapour originated from the local evaporation over quiescent oceanic conditions.
Depending on the atmospheric circulation, on deep convective and mixing processes and on
the site of interest, the initial site may be characterised by a different isotopic composition:
$R_{vi} = R_{v0} + \delta R_{vi}$ (3)
The residual fraction $f$ depends on the minimum condensation temperature that the parcel has
undergone along its trajectory towards the site of interest, $T^*$ (Galewsky and Hurley, 2010;
Galewsky et al., 2005; Sherwood, 1996):
$f = q_s(T^*)/q0$ (4)
where $q_s$ is the saturation specific humidity, function of temperature following the Clausius-
Clapeyron relationship.



If we assume that the air at the site of interest has been transported adiabatically from the area
of minimum condensation temperature, then:
$q_s(T^*) = h \cdot q_s(T_s)$ (5)
when $h$ and $T_s$ are the relative humidity and air temperature near the surface of the site of
interest.
The surface temperature can be predicted to the first order by the adiabatic lapse rate, $\Gamma$, and
is modulated by the non-adiabatic component, $\delta T_s$ that represents processes such as large-
scale circulation or radiation:
$T_s = T_0 + \Gamma \cdot (z - z_o) + \delta T_s$ (6)
where $z$ and $z_0$ are the altitudes at the site of interest and at the initial site. We use an adiabatic
lapse rate equal to 5° km$^{-1}$ based on the measurements of modern observed mean temperature
lapse rate on the southern slope of the central Himalayas, that ranges from 4.7 to 6.1° km$^{-1}$ for
the monsoon season and from 4.3 to 5.5° km$^{-1}$ for the rest of the year (Kattel et al., 2015).
If we combine Eq. (1) to Eq. (6), we get that $R_v$ is a function of $\delta R_{vi}$, $\varepsilon$, $h$, $\delta T_s$ and $z$:
$R_p = R_p(\delta R_{vi}, \varepsilon, h, \delta T_s, z)$ (7)
Parameters $z_0$, $q_0$, $T_0$ are reference values that are common to all sites of interest, all climates
and geographies. Even if initial conditions for the Rayleigh distillation vary depending on the
atmosphere circulation, on deep convective processes and on the site of interest, we keep the
same reference values and we consider all variations in initial conditions are accommodated
by $\delta R_{vi}$.
This model is equivalent to that of Rowley et al. (2001) for $\delta R_{vi} = 0$ (i.e. neglecting the effects
of mixing and deep convection on the initial water vapour), $\varepsilon = (\alpha - 1) \cdot R_v$ (i.e. neglecting
post-condensational effects), and $h = 1$ (i.e. assuming the site of interest is inside the
precipitating cloud).

## 2.3 Decomposing precipitation composition differences

Our goal is to understand why $R_p$ varies from one climatic state to another. Let's refer to these
climatic states using subscript 1 and 2 and to their difference using the $\Delta$ notation.
Differences between INT and LOW and between MOD and INT climatic states corresponds
to the initial and the terminate stages of the TP uplift respectively. We decompose $\Delta R_p = R_{p2} -$
$R_{p1}$ into contribution from $\Delta \delta R_{vi}$, $\Delta \varepsilon$, $\Delta h$, $\Delta \delta T_s$, and $\Delta z$:



$$\Delta R_p = \frac{\partial R_p}{\partial R_{vi}} \cdot \Delta \delta R_{vi} + \frac{\partial R_p}{\partial \varepsilon} \cdot \Delta \varepsilon + \frac{\partial R_p}{\partial h} \cdot \Delta h + \frac{\partial R_p}{\partial \delta T_s} \cdot \Delta \delta T_s + \frac{\partial R_p}{\partial z} \cdot \Delta z \qquad (8)$$
To estimate each of these terms, we estimate difference between $R_p$ calculated from the
different values of $\delta R_{vi}$, $\varepsilon$, $h$, $\delta T_s$, and $z$, changing only one parameter at a time, as detailed in
table 1 (and see next section). Our method to estimate the terms in Eq. (8) is equivalent to first
order approximation of partial derivatives, i.e. we neglect the sensitivity of the partial
derivatives to the state at which they are calculated.
Values of $\delta R_{vi}$, $\varepsilon$, $h$, $\delta T_s$, and $z$, are diagnosed using LMDZ-iso simulations. As an example
$R_p(\delta R_{vi2}, \varepsilon_2, h_2, \delta T_{s2}, z_2)$ is the precipitation composition simulated by LMDZ for climate
state 2. As another example, $R_p(0,0,1, \delta T_{s1}, z_1)$ is the precipitation composition predicted by
Eqs. (2)-(5) with $\delta R_{vi} = 0$ and using the near-surface air temperature as $T_s$ simulated by
LMDZ for climatic state 1 (see Table 1).
**3   Results**
**3.1   Impact of TP uplift on Asian climate**
Theoretically, the Tibetan Plateau has both mechanical and thermal effects on atmospheric
dynamics that induce increase monsoon activity to the south and drive arid climate to the
north (Broccoli and Manabe, 1992; Sato and Kimura, 2005). Thus modifying TP height is
expected to alter these large-scale atmospheric dynamics and associated climate variables
(namely temperature, precipitation, relative humidity (hereafter RH), cloud cover), and in turn
to affect the isotopic signature of rainfall.
In LOW experiment, strong summer heating leads to the onset of a "Thermal Low" (TL) at
the latitude of maximal insolation (ca. 32°N), similar to the present-day TL existing over the
Sahara desert (Fig. S2). This structure is superimposed by large-scale subsidence linked to the
descending branch of the Hadley cell, and both factors act to drive widespread aridity over TP
area between ca 30°N and 40°N, associated with very low (<40%) RH values (Fig. S2).
Subsidence also prevents the development of South Asian monsoon over the north Indian
plane and favours aridity over this region. In winter, large-scale subsidence induces high
surface pressures and creates a anticyclonic cell that prevents convection and humidity
advection, resulting in low RH and annual rainfall amount ranging from 50 to 500 mm over
TP area (Fig. 3).



Uplifting TP from 250m above sea-level (ASL) to half of its present-day altitude (INT case)
initiates convection in the first tropospheric layers, restraining large-scale subsidence to the
upper levels (Fig. 3). In turn, south Asian monsoon is strengthened and associated northward
moisture transport and precipitation increase south of TP (Fig. 4, 5). As a consequence the
hydrological cycle over TP is more active, with higher evaporation rates (Fig. 6 D). Together
with colder temperatures linked to higher altitude (adiabatic effect) (Fig. 6 B), the stronger
hydrological cycle drives an increase in RH (Fig. 6 A) and cloud cover (Fig. S3). Another
consequence of increased altitude is higher snowfall rates in winter and associated rise of
surface albedo (fig. S4). When added to the increased cloud cover effect, this last process
contributes to an extra cooling of air masses over the Plateau. To the north of TP, the initial
stage of uplift results in increased aridity (i.e. lower RH and rainfall) over the Tarim Basin
region. This pattern can be explained both by a barrier effect of southern topography and by
stationary waves strengthening, that results in subsidence to the north of TP. This latter
mechanism is consistent with pioneer studies which showed that mountain-related activation
of stationary waves prevented cyclonic activity over Central Asia and induced aridity over
this region (Broccoli and Manabe, 1992).
The impact of the terminal stage of TP uplift also drives an increase in RH over the Plateau,
especially during summer time, when a very active continental recycling (Fig. S5) makes RH
rise from 40% (INT) to 70% (MOD). Precipitation amount also increases significantly (Fig.
5), driven both by increased evaporation and water recycling during summer, and intense
snowfall during winter. The latter contributes to increase the surface albedo and associated
surface cooling during winter. Conversely, the uplift to a modern-like Plateau reduces RH
(down to 30%) north of the Plateau, and allows the onset of large arid areas. We infer that this
aridification is linked to a mechanical blocking of moisture transport, both by Tian Shan
topography for the winter westerlies, and the eastern flanks of TP for summer fluxes, since
despite changes in stationary waves structure and sensible heat (not shown), no marked shift
in subsidence between INT and MOD experiments is simulated. This result is consistent with
recent studies (Miao et al., 2012; Sun et al., 2009) that have suggested the potential
contribution of Pamir and Tian Shan rainshadow effect to aridification in Quad Basin and
creation of Taklamakan Desert.



## 3.2 Response of precipitation $\delta^{18}$O to TP uplift
### 3.2.1 Model validation
The modern mean annual isotopic distribution is characterised by very depleted values of
$\delta^{18}$O over the Himalayas and the southern Tibet (down to -18‰) and a shift to more positive
values (ranges from -11 to -13‰) over northern TP and Kunlun from 30°N to 35°N.
Precipitation $\delta^{18}$O over Tarim Basin experiences an abrupt decrease compared to northern TP,
with values down to -16‰. (Fig. 7 A). Overall, simulated annual mean $\delta^{18}$O$_p$ are consistent
with sparse observations from the International Atomic Energy Agency (IAEA) Global
Network of Isotopes in Precipitation and $\delta^{18}$O in precipitation measurements compiled from
Caves et al. (2015) (Fig. 7 A). In general, model shows a good agreement with precipitation
and VSMOW-weighted modern surface waters $\delta^{18}$O, including stream, lake and spring waters
(data from Bershaw et al., 2012; Hren et al., 2009; Quade et al., 2011). This comparison
shows ability of our model to reproduce decrease in $\delta^{18}$O from India subcontinent to
Himalayas foothills and with minimum values over the Himalayas. Simulated increase in $\delta^{18}$O
over the TP with the distance from the Himalayas is consistent with data sampled along a
southwest-northeast transect across the Plateau (Bershaw et al., 2012). Model-data
discrepancies occur over central Tibetan Plateau where measured data have extremely
positive values probably linked with surface processes including high recycling rate and
contamination of streams with groundwater, that shifts surface water $\delta^{18}$O to more positive
values compared to those in precipitation (Bershaw et al., 2012).
### 3.2.2 Simulated isotopic changes and signal decomposition
To first order, increasing topography over TP leads to more negative $\delta^{18}$O over the region
(Fig. 7). In the absence of topography, precipitation $\delta^{18}$O follows a zonal pattern and
undergoes a weak latitudinal depletion on the way to the continental interior, except from
slight deviations over the Indian plane, central China and the Eastern part of the TP (Fig. 7 C).
At 40°N, i.e. the northern edge of modern TP, $\delta^{18}$O values reaches -9‰ in LOW case,
compared to -14‰ in MOD case. For the INT case the latitudinal depletion from south to
north is stronger (ca. 0.4‰ per latitudinal degree), with $\delta^{18}$O values ranging from -6‰ for the
lowered Himalayas foothills to -11‰ for northern and eastern margins of TP (Fig. 7 B).



The total difference in isotopic composition of precipitation, $\Delta R_p$, between experiments (INT-
LOW, MOD-INT) is significant beyond the areas where the topography was reduced by the
experimental design (Fig. 8 A, Fig. 9 A). Substantial differences in $\delta^{18}O$ between MOD and
INT experiments are simulated over the southern TP (up to 10‰) and over the Tarim Basin
(up to 7‰). Between INT and LOW cases, the differences are over the margins of the TP,
over Pamir, Tian Shan and Nan Chan. We should note that the isotopic difference becomes
more important for the later stage of the plateau uplift.  For clarity, we define two boxes, over
the northern (from 34°N to 38°N and from 88°E to 100°E) and southern (from 27°N to 33°N
and from 75°E to 95°E) part of TP.
**Direct topography effect on $\delta^{18}O$**
The direct effect of topography change is determined as the decomposition term $\frac{\partial R_p}{\partial z} \cdot \Delta z$ in
Eq. (8). For the initial stage of the uplift, the altitude effect produces a decrease in
precipitation $\delta^{18}O$ ranging from -1 to -3‰ (Fig. 8 B). For the terminal stage of the uplift, the
isotopic decrease linked with altitude goes up to -7‰ (Fig. 9 B). Differences between both
stages are linked to the non-linear relationship between $\delta^{18}O$ and elevation. Also for both
stages, the difference between $\Delta Rp$ and $\frac{\partial R_p}{\partial z} \cdot \Delta z$  is non-zero (Fig. 10 A, Fig. 11 A). These
differences are particularly marked for the terminal stage, for which $\frac{\partial R_p}{\partial z} \cdot \Delta z$ averages -5.5‰
over the northern part of TP (Fig. 11 A B), whereas the total isotopic change averages -3‰.
Locally, the difference between $\frac{\partial R_p}{\partial z} \cdot \Delta z$ and $\Delta R_p$ can reach +4‰. When averaged over the
southern box, $\frac{\partial R_p}{\partial z} \cdot \Delta z$ is less negative (-4‰) than $\Delta Rp$ (-4.6‰), with localized maximum
differences reaching -4‰. Offsets between $\frac{\partial R_p}{\partial z} \cdot \Delta z$ and $\Delta Rp$ are also detected for the initial
stage of the uplift (Fig. 10 A B), but are lower: they reach +2‰ over central TP but barely
reach 1‰ when averaged over southern and northern boxes. These offsets are related to
additional effects of uplift on $\delta^{18}O$ that are discussed in the following sections.
**Non-adiabatic temperature changes impact**
Besides the adiabatic temperature effects linked with the TP uplift, non-adiabatic temperature
changes can be identified, in relation with surface albedo and cloud cover changes depicted in



3.2.1. The term $\frac{\partial R_p}{\partial \delta T_s} \cdot \Delta \delta T_s$ in Eq. (8) (Table 1, line 3) is associated with these non-adiabatic
effects, i.e. spatial variations of the temperature lapse rate. Figure 8 C and Figure 9 C show
the portion of the total isotopic signal that is linked to this effect. It plays a modest role for the
early phase of uplift (+1-2‰ locally), but is more important for the second stage. It
contributes to 2-3‰ of total isotopic difference, with a positive sign over southeast TP
interior, TP northern margins and Asia interior. Negative anomalies have the same magnitude,
but are less widespread, localized over the TP interior (Fig. 19 C). Positive isotopic anomalies
are associated with steeper lapse rate than expected based on adiabatic processes. Conversely,
negative $\delta^{18}$O anomalies that are observed over northern TP and over Pamir are explained by
a weaker lapse rate than adiabatic. Overall, these variations represent between 7 and 15% (2-
8% for the initial stage) of the processes that are not linked to topography (Fig. 10 D, E and
11 D, E).
**Impact of RH changes during condensation process**
The term $\frac{\partial R_p}{\partial h} \cdot \Delta h$ in Eq. (8) depicts the portion of total isotopic signal $\Delta R_p$ linked to local RH
change during condensation process (Table. 1, line 4). Over TP, $\frac{\partial R_p}{\partial h} \cdot \Delta h$ is positive for both
uplift phases, and RH changes act as a counterbalance to the topography effect. $\frac{\partial R_p}{\partial h} \cdot \Delta h$
reaches +6‰ for the late stage (Fig. 9 D), and maxima are located over western part and
northern part of TP for both stages of the uplift. Equation (5) shows that this positive anomaly
is directly related to the increase in RH described in 3.2.1. For the initial stage, $\frac{\partial R_p}{\partial h} \cdot \Delta h$
depicts also positive values (up to +4‰) to the southwest of TP. When averaged over
northern and southern boxes, the counterbalancing effect of RH on $\Delta Rp$ ranges from 1.5 to
+3‰, and this effect represents up to 76% of all non-topographic processes (Fig. 10, 11).
Interestingly, an opposite signal is simulated over the Tarim basin, where topography was
kept constant in the three experiments. This signal is consistent with the previously-depicted
decrease in RH over this region, in relation with rain-shadow effects and large-scale
subsidence.
**Post-condensation processes impact**





Estimation of term $\frac{\partial R_p}{\partial \varepsilon} \cdot \Delta\varepsilon$, i.e. the change in isotopic difference between vapour and
precipitation, allows to quantify the contribution of post-condensational processes to total
$\Delta R_p$ signal (Fig. 8 E, 9 E). For both stages of uplift, $\frac{\partial R_p}{\partial \varepsilon} \cdot \Delta\varepsilon$ is mostly negative, indicating a
depletion of $R_p$ relatively to $R_v$ with the uplift. Over the Plateau, contribution of post-
condensational effects for the initial stage of uplift ranges from 25% to 46% of total non-
topographic effects, whereas it represents less than 10% for the terminal stage (Fig. 10 A, 11
A). The most significant signal is simulated over the northern part of the Plateau and over its
western margin and adjacent areas. Post-condensational effects during the initial stage lead to
up to a -5‰ anomaly over the western margin of TP (Fig. 10 E) whereas the terminal stage
creates a substantial negative anomaly only over northern TP margin and Tarim Basin
(Fig. 11 E).
**Residual processes effect**
The last term of Eq. (8), $\frac{\partial R_p}{\partial R_{vi}} \cdot \Delta\delta R_{vi}$, corresponds to the part of the total isotopic signal that
could not be explained by previously mentioned processes. These residual anomalies are
rather weak for the initial stage of the uplift, explaining less than 1‰ of the signal over the
northern plateau, and around 1‰ over the southern TP and adjacent parts of Asia and India
(Fig. 8 F). Contribution of these effects to the initial stage is 4% and 21% to the northern and
southern box respectively (Fig. 10 D E). Conversely, for the terminal stage of the TP uplift
this anomaly reaches up to -4‰ over the southern part of the TP (Fig. 9 F) and contributes to
49% of the non-topographic processes signal (Fig. 11 D E). In the next sections we propose
several mechanisms that could contribute to this residual anomaly.
**4   Discussion**
Our results suggest that TP uplift affects precipitation $\delta^{18}O$ through direct topographic effect,
but that a significant part of the signal is related to several other processes. These processes
alter the isotopic signal not only over TP, but also over adjacent regions, where topography
was kept the same by the experiment design. A second result is that despite a similar
altitudinal change of TP between the two uplift stages, the topographic effect on $\delta^{18}O$ is more
perturbed by other processes during the terminal stage than during the initial one.



For the terminal stage, the residual effects change over the southern region dominates (49%)
the isotopic signal that is not linked to the direct topographic effect. The RH change and non-
adiabatic temperature changes also have an important counterbalancing impact, together
contributing to 43% of the isotopic signal (Fig. 11 E). For the northern region, the topographic
effect is mainly counterbalanced by the RH change effect (2.5‰), ultimately leading to a
2.3‰ offset between $\Delta R_p$ and what expected from topography. Here RH contributes to 76%
of the isotopic signal not linked with the topography change, while non-adiabatic temperature
changes, residual effects change and post-condensational processes have at impact of 16%,
7% and <1% respectively (Fig. 11 D).

## 11  4.1  Impact of RH variations

RH alters rainfall isotopic signature through two steps, during and after condensation. As
mentioned earlier, the first effect of RH, as shown in Eq. (5) and expressed as $\frac{\partial R_p}{\partial h} \cdot \Delta h$, occurs
during condensation through Rayleigh distillation and induces that $R_p$ increases with
increasing RH. Our model shows that RH increases over TP with the initial stage of uplift,
driving precipitation $\delta^{18}O$ towards less negative values. This mechanism is more efficient for
the terminal stage of uplift, when RH is increased in summer as a response of a more active
water cycle. South of TP, RH direct effect on $\delta^{18}O$ is noticeable, as efficient moisture
transport is activated with the uplift-driven strengthening of monsoon circulation (Fig. 4).
Interestingly, this mechanism is not active for the second stage of the uplift, during which
rainfall increases through more effective convection, not through higher advection of
moisture. As a consequence, negligible RH and $R_p$ changes are simulated south of the Plateau
when it reaches its full height. This suggests that an altitudinal threshold might trigger south
Asian monsoon strengthening, and ultimately precipitation $\delta^{18}O$ signature, a hypothesis that
should be explored in further studies. Conversely, the negative values of $\frac{\partial R_p}{\partial h} \cdot \Delta h$ over and
northeast of the Tarim basin are related to a decrease in RH during both stages. Our analysis
suggests that the first uplift stage is sufficient to create both barrier effects to moisture fluxes
and large-scale subsidence that ultimately drive aridity over the region.
The second effect of RH on $\delta^{18}O$ concerns very dry areas (ca. < 40%), where raindrop re-
evaporation can occur after initial condensation, leading to an isotopic enrichment of
precipitation compared to water vapour (Lee and Fung, 2008) (Fig. S2). Such an effect is





implicitly included in the post-condensational term of our decomposition that shows opposite
sign when compared to $\frac{\partial R_p}{\partial h} \cdot \Delta h$. Over the Plateau, this mechanism is effective only for the
first uplift stage, where TP area transits from very low precipitation amounts and very low RH
values to wetter conditions (Fig. S6).
Over TP, the opposed effects of RH almost compensate each other for the early stage of the
uplift (Fig. 8 D, E), but it is not the case for the final stage, since RH post-condensational
effect is similar between INT and MOD experiments. Since absolute values of the impact of
RH through condensation and post-condensational processes can reach 5‰, it is crucial to
consider RH variation when inferring paleoaltitudes from carbonates $\delta^{18}$O.

## 11   4.2   "Amount effect" and monsoon intensification

Our results also show a substantial increase in precipitation amount over northern India, the
Himalayas and TP with the growth of topography for both uplift stages (Fig. 12). The inverse
relation between the enrichment in heavy isotopes in precipitation and precipitation amount,
named the "amount effect" (Dansgaard, 1964) is largely known for oceanic tropical
conditions (Risi et al., 2008; Rozanski, Kazimierz Araguás-Araguás and Gonfiantini, 1993)
and for Asia monsoonal areas (Lee et al., 2012; Yang et al., 2011). Over South Tibet recent
studies have shown the role of deep convection in isotopic depletion (He et al., 2015). For the
two stages of uplift, the residual component of the isotopic signal depicts negative values over
southern TP, where annual rainfall amount is increased. Thus we infer that this anomaly can
be driven, at least partly, by the amount effect that increases with growing topography.
Various climate studies have suggested that the appearance of the monsoonal system in East
Asia and the onset of central Asian desertification were related to Cenozoic Himalayan–
Tibetan uplift and withdrawal of the Paratethys Sea (Clift et al., 2008; Guo et al., 2002, 2008;
Kutzbach et al., 1989, 1993; Ramstein et al., 1997; Raymo and Ruddiman, 1992; Ruddiman
and Kutzbach, 1989; Sun and Wang, 2005; Zhang et al., 2007; Zhisheng et al., 2001) although
the exact timing of the monsoon onset and its intensification remains debated (Licht et al.,
2014; Molnar et al., 2010). Although our experimental setup, which does not include
Cenozoic paleogeography, was not designed to assess the question of monsoon driving
mechanisms nor its timing, our results suggest that uplifting the Plateau from 250 meters ASL
to half of its present height is enough to enhance moisture transport towards northern India





and strengthen seasonal rainfall. Nevertheless, massive increase of rainfall over TP between
INT and MOD experiments indicates that the second phase of uplift might be crucial to
activate an efficient, modern-day-like, hydrological cycle over the Plateau. The decrease in
simulated precipitation north of the Plateau also suggests that terminal phase of TP uplift
triggered modern-day arid areas.

## 7 4.3 Other effects

Although precipitation amount change explains well the residual isotopic anomaly (Fig. 8 F,
Fig. 9 F), additional processes could interplay. Continental recycling can overprint original
moisture signature and shifts the isotopic ratios to higher values due to recharging of moisture
by heavy isotopes from soil evaporation (Lee et al., 2012; Risi et al., 2013). In our simulation,
we detect an increasing role of continental recycling in the hydrological budget of the TP
(Fig. S5), especially in its central part, that likely shifts the $\delta^{18}$O to more positive values and
partially compensate for the depletion linked to the "amount effect" over the central plateau.
Another process frequently invoked to explain the evolution of precipitation $\delta^{18}$O patterns
over TP is changes in moisture sources (Bershaw et al., 2012; Dettman et al., 2003; Quade et
al., 2007; Tian et al., 2007). Except for the continentally recycled moisture, southern
Himalayas precipitation moisture originates mainly from the Indian, the Atlantic and the
Pacific Oceans (Fig. S5). Proximate oceanic basins are known to be sources of moisture with
more positive signature than remote ones (Chen et al., 2012; Gat, 1996). Supplemental
analyses with water-tagging feature of LMDZiso show that contribution of continental
recycling to rainfall over TP increases with the uplift, at the expanse of Pacific and Indian
sources (Fig. S5). Although we have no mean to decipher between sources and amount effect
in the residual anomaly, it seems that the change of sources is not sufficient to yield a strong
offset of $\delta^{18}$O values.

## 27 4.4 Relevance of paleoelevation reconstructions based on paleo $\delta^{18}$O

Quantitative paleoelevation reconstructions using modern altitude-$\delta^{18}$O relationship will
succeed only if $\Delta R_p$ corresponds mainly to the direct topography effect. Modern
paleoaltimetry studies cover almost all regions of the Plateau for time periods ranging from
Palaeocene to Pleistocene-Quaternary (see data compilation in Caves et al., 2015). Most of





these studies consider changes in $\delta^{18}O$ as a direct effect of the topography uplift.
Paleoelevation studies locations (see Caves et al., 2015 for a synthesis) plotted over the
anomaly maps (Fig. 10 A, Fig. 11 A) show for what geographical regions restored elevations
should be used with an additional caution. Numerous paleoelevation data points were located
either over the northern part of the TP (from 34°N to 38°N and from 88°E to 100°E) or over
the southern region (from 27°N to 33°N and from 75°E to 95°E).
Our model results show that when TP altitude is increased from half to full, considering
topography as an exclusive controlling factor of precipitation $\delta^{18}O$ over the southern
(northern) region likely yield overestimations (underestimations) of surface uplift, since the
topography effect is offset by RH and amount effects. Projecting our modelling results to each
locality where paleoelevation studies have been published (Table 3) reveals that topography
change explains simulated total isotopic change reasonably well for only few locations
(Linzhou Basin, Lunpola Basin, Kailas Basin, Huaitoutala).  Indeed topography appears to be
the main controlling factor for only 40% of the sites, while 30% are dominated by RH effects,
13% by residual effects and 5% and 2% by post-condensational and non-adiabatic
temperature changes, respectively. Nevertheless such figures have to be taken carefully, since
we ran idealized experiments testing only the impact of uplift, neglecting other factors like
horizontal paleogeography or $pCO_2$ variations, the latter being known to influence $\delta^{18}O$ as
well (Jeffery et al., 2012; Poulsen and Jeffery, 2011).
For the initial uplift stage apparent consistency occurs between the topography impact and the
total isotopic composition is observed, in relation with counteracting effects RH and pots-
condensational processes. For the southern region RH impact is appeared to be the main
controlling factor for the isotopic composition of precipitation, surpassing the direct
topography impact. Nevertheless, these processes have a different contribution for initial and
terminal stages of uplift. Precipitation changes lead to overestimate altitude changes for both
stages, but for the terminal stage its contribution is bigger. This effect dominates in the
southern part, and more generally where the isotopic composition of precipitation strongly
depends on convective activity. RH changes dominate over the western part of TP and
Northern India for initial uplift stage and over the northern TP for the terminal.



## 5   Conclusions

Previous studies focusing on the Andes (Ehlers and Poulsen, 2009; Poulsen et al., 2010) or north American cordillera (Sewall and Fricke, 2013) have inferred that the impact of uplift of mountain ranges on $\delta^{18}O$ could be altered by the consequences of the uplift on atmospheric physics and dynamics. Our modelling results show that it is also the case for the Tibetan Plateau uplift. Additionally, we designed a decomposing analysis to quantify for the first time the different processes that can alter precipitation $\delta^{18}O$ changes with uplift. As suggested for the Andes, the onset of convective rainfall plays an important role in shifting $\delta^{18}O$ towards more negative values. Nevertheless this process is not the main factor, as we show that saturation of air masses, quantified by RH have two to three-time bigger effects on the final $\delta^{18}O$. We infer that increase in precipitation linked with the TP uplift would lead to overestimation of the topography uplift at sites over Himalayas and Southern TP, whereas increase in RH leads to underestimating the uplift at sites in Northern Tibet.

Our results could be applied to interpret paleoclimate records and to reconstruct the region uplift history. Paleoelevation reconstructions suggest the Himalayas attained their current elevation by the late Miocene (Garzione et al., 2000a, 2000b; Rowley et al., 2001; Saylor et al., 2009). Our results show overestimation of the topography impact over this region, thus the Himalayas may have attained their current elevation later than expected. In contrast, isotope-based paleoaltimetry could underestimate surface elevation over the northern TP. This could explain why available isotope-based paleoelevation estimates for the northern TP (Cyr et al., 2005), which estimates surface elevation about 2km, contradict palynological assemblages in lacustrine sediments from the Xining basin, which show the presence of high-altitude vegetation at the same time period (Dupont-Nivet et al., 2008; Hoorn et al., 2012).

Still, our decomposition methods reveal that even if the impact of the TP uplift phases are rather straightforward (monsoon enhancement to the South, increase in continental recycling over TP, moisture fluxes deflection and increased aridity to the North), the consequences in terms of $\delta^{18}O$ are extremely complex, since interplays and compensation occur amongst all the processes. Limitations in our approach are related to 1/ the theoretical uplift scenario we chose, and 2/ the fact that all other boundary conditions are set to present. Concerning the latter, changes in vegetation cover, by altering albedo and persistence of snow cover, could affect the impact of non-adiabatic temperature changes on $\delta^{18}O$. Vegetation over Asia was shown to have a major variation through Cenozoic based on pollen (Dupont-Nivet et al.,



2008; Miao et al., 2011; Song et al., 2010; Zhao and Yu, 2012) and paleobolanical data  (An
et al., 2005; De Franceschi et al., 2008; Kohn, 2010) and future studies would benefit to
explore its impact on precipitation $\delta^{18}$O. Also it is largely known that during the Cenozoic air
temperature was higher due to higher concentration of greenhouse gases in the atmosphere
(Zachos et al., 2008). Studies taking into account this feedback inferred that it could lead to
even larger inaccuracy in surface uplift estimations during the Cenozoic (Poulsen and Jeffery,
2011). Thus the field of paleoaltimetry would benefit from future studies applying a
decomposition method and comparing paleoaltitude data to GCM simulations forced by
constrained paleogeography (land-sea mask and different scenarios for orogens) and
atmospheric $pCO_2$.

## Acknowledgements

This work is a part of iTECC (interaction Tectonics-Erosion-Climate-Coupling) project
funded by European Union. Computational resources were provided by IDRIS-GENCI
(project 0292), France

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



1   Table 1. Table detailing how the different terms of the decomposition for $\Delta R_p$, as written in

2   Eq. (8), are estimated

| Term written with differential format | Estimate of these terms | Physical meaning |
|---|---|---|
| $\Delta R_p$ | $R_p(\delta R_{vi2}, \varepsilon_2, h_2, \delta T_{s2}, z_2)$ $- R_p(\delta R_{vi1}, \varepsilon_1, h_1, \delta T_{s1}, z_1)$ | Total isotopic difference between state 1 and state 2 |
| $\dfrac{\partial R_p}{\partial z} \cdot \Delta z$ | $R_p(0,0,1,0,z_2) - R_p(0,0,1,0,z_1)$ | Direct effect of topography change |
| $\dfrac{\partial R_p}{\partial \delta T_s} \cdot \Delta \delta T_s$ | $R_p(0,0,1,\delta T_{s2}, z_2) - R_p(0,0,1,\delta T_{s1}, z_1)$ $-(R_p(0,0,1,0,z_2) - R_p(0,0,1,0,z_1))$ | Effect of lapse rate change, associated with non-adiabatic effects, possibly due to changes in surface energy budget or in large-scale atmospheric stratification |
| $\dfrac{\partial R_p}{\partial h} \cdot \Delta h$ | $R_p(0,0,h_2,\delta T_{s2}, z_2) - R_p(0,0,h_1,\delta T_{s1}, z_1)$ $-(R_p(0,0,1,\delta T_{s2}, z_2) - R_p(0,0,1,\delta T_{s1}, z_1))$ | Effect of local relative humidity change, possibly due to large-scale circulation changes |
| $\dfrac{\partial R_p}{\partial \varepsilon} \cdot \Delta \varepsilon$ | $R_p(0,\varepsilon_2, h_2, \delta T_{s2}, z_2)$ $- R_p(0,\varepsilon_1, h_1, \delta T_{s1}, z_1)$ $-(R_p(0,0,h_2,\delta T_{s2}, z_2)$ $- R_p(0,0,h_1,\delta T_{s1}, z_1))$ | Effect of changes in condensational and post-condensational effects, possibly due to changes in rain reevaporation processes |
| $\dfrac{\partial R_p}{\partial R_{vi}} \cdot \Delta \delta R_{vi}$ | $R_p(\delta R_{vi2}, \varepsilon_2, h_2, \delta T_{s2}, z_2)$ $- R_p(\delta R_{vi1}, \varepsilon_1, h_1, \delta T_{s1}, z_1)$ | All other effects, including effects of deep convection, mixing, water vapor origin, |





| | |
|---|---|
| $-(R_p(0, \varepsilon_2, h_2, \delta T_{s2}, z_2)$ $- R_p(0, \varepsilon_1, h_1, \delta T_{s1}, z_1))$ | continental recycling on the initial water vapor |



1    Table 2. Values of isotopic changes due to decomposed terms for two uplift stages and for

2    two regions (see the text)

| Term | Isotopic change (‰) | | | |
|---|---|---|---|---|
| | Initial Stage | | Terminal Stage | |
| | South | North | South | North |
| $\frac{\partial R_p}{\partial z} \cdot \Delta z$ | -1.40 | -2.00 | -3.96 | -5.50 |
| $\frac{\partial R_p}{\partial \delta T_s} \cdot \Delta \delta T_s$ | 0.4 | -0.09 | 0.76 | -0.25 |
| $\frac{\partial R_p}{\partial h} \cdot \Delta h$ | 2.40 | 1.97 | 1.38 | 2.50 |
| $\frac{\partial R_p}{\partial \varepsilon} \cdot \Delta \varepsilon$ | -1.30 | -1.73 | -0.41 | 0.01 |
| $\frac{\partial R_p}{\partial R_{vi}} \cdot \Delta \delta R_{vi}$ | -1.10 | -0.14 | -2.38 | -0.54 |
| Total $\Delta R_p$ | -1.00 | -1.99 | -4.61 | -3.16 |



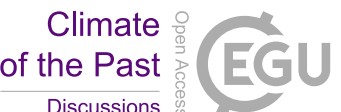

2    Table 3. Impact of the different terms of the decomposition on the isotopic signal for the

3    terminal stage of HTP uplift in the location where paleoelevation studies have been done

| Locality | Latitude | Longitude | $\Delta R_p$ (‰) | $\frac{\partial R_p}{\partial z} \cdot \Delta z$ (‰) | $\frac{\partial R_p}{\partial \delta T_s} \cdot \Delta \delta T_s$ (‰) | $\frac{\partial R_p}{\partial \varepsilon} \cdot \Delta \varepsilon$ (‰) | $\frac{\partial R_p}{\partial h} \cdot \Delta h$ (‰) | $\frac{\partial R_p}{\partial R_{vi}} \cdot \Delta \delta R_{vi}$ (‰) | Paleoelevation studies at this locality |
|---|---|---|---|---|---|---|---|---|---|
| Aertashi | 37.97 | 75.55 | -1.619 | -2.859 | 0.9986 | -0.2935 | -0.2679 | 0.8033 | Kent-Corson et al. (2009) |
| Biger Noor | 45.90 | 96.78 | -1.169 | -0.004360 | 2.673 | -0.7024 | -4.419 | 1.283 | Caves et al. (2014) |
| Chake Basin | 23.80 | 103.10 | -0.2520 | 0.006192 | -0.03002 | 0.04205 | 0.2631 | -0.5333 | Hoke et al. (2014) |
| Dzereg | 47.14 | 93.06 | -1.006 | -0.003550 | 2.216 | -0.3723 | -4.313 | 1.466 | Caves et al. (2014) |
| Eryuan | 26.20 | 99.80 | -1.356 | -1.574 | 0.6337 | 0.1705 | 0.4971 | -1.083 | Hoke et al. (2014) |
| Ganchaigou | 37.69 | 91.04 | -3.195 | -2.780 | 0.8363 | -1.292 | 0.6104 | -0.5692 | Kent-Corson et al. (2009) |
| Gyirong Basin | 28.70 | 85.25 | -7.017 | -3.850 | 1.073 | -1.089 | 0.4085 | -3.559 | Wang et al. (1996) |
| Hexi Corridor | 39.52 | 97.52 | -2.907 | -0.2788 | 1.732 | -1.985 | -2.293 | -0.08312 | Kent-Corson et al. (2009) |
| HohXil Basin | 34.60 | 93.00 | -3.972 | -6.529 | 0.6597 | 0.03659 | 3.375 | -1.514 | Cyr et al. (2005) |
| Huaitoutala | 37.30 | 96.70 | -5.998 | -4.418 | -1.473 | -3.620 | 3.104 | 0.4089 | Zhuang et al. (2011) |
| India Siwaliks | 30.35 | 77.60 | -1.862 | 0.005774 | 0.1029 | -0.3033 | 0.1828 | -1.851 | Ghosh et al. (2004) |
| India Siwaliks | 30.34 | 77.60 | -1.862 | 0.005774 | 0.1029 | -0.3033 | 0.1828 | -1.851 | Sanyal et al. (2005) |
| Janggalsay | 38.15 | 86.62 | -4.487 | -2.406 | 1.026 | -2.347 | -0.9517 | 0.1923 | Kent-Corson et al. (2009) |





| | | | | | | | | |
|---|---|---|---|---|---|---|---|---|
| Jianchuan Basin | 26.60 | 99.80 | -1.356 | -1.574 | 0.6337 | 0.1705 | 0.4971 | -1.083 | Hoke et al. (2014) |
| Jingou | 44.75 | 85.40 | 1.073 | -0.03134 | 1.270 | 1.435 | -2.054 | 0.4526 | Charreau et al. (2012) |
| Kailas Basin | 31.20 | 81.00 | -6.705 | -7.181 | 0.4011 | 0.7988 | 3.162 | -3.886 | DeCelles et al. (2011) |
| Kuitun | 45.00 | 84.75 | 1.073 | -0.03134 | 1.270 | 1.435 | -2.054 | 0.4526 | Charreau et al. (2012) |
| Lake Mahai | 37.66 | 94.24 | -0.9639 | -0.002379 | 2.737 | 0.4231 | -4.188 | 0.06643 | Kent-Corson et al. (2009) |
| Lanping | 26.50 | 99.40 | -1.356 | -1.574 | 0.6337 | 0.1705 | 0.4971 | -1.083 | Hoke et al. (2014) |
| Lao Mangnai | 36.94 | 91.96 | -1.133 | -3.998 | 0.4470 | 0.3556 | 2.233 | -0.1708 | Kent-Corson et al. (2009) |
| Lenghu | 37.84 | 93.36 | -0.9639 | -0.002379 | 2.737 | 0.4231 | -4.188 | 0.06643 | Kent-Corson et al. (2009) |
| Linxia Basin | 35.69 | 103.10 | 0.4433 | -0.9613 | 1.079 | 0.3638 | -0.4095 | 0.3712 | Dettman et al. (2003) |
| Linzhou Basin | 30.00 | 91.20 | -6.756 | -5.956 | 2.337 | -0.05738 | 0.8857 | -3.965 | Ding et al. (2014) |
| Luhe | 25.20 | 101.30 | -0.2417 | 0.008419 | 0.3172 | 0.4105 | -0.2359 | -0.7419 | Hoke et al. (2014) |
| Lulehe | 37.50 | 95.08 | -0.06084 | -0.9874 | 1.724 | 1.950 | -3.326 | 0.5784 | Kent-Corson et al. (2009) |
| Lulehe | 37.50 | 95.08 | -0.06084 | -0.9874 | 1.724 | 1.950 | -3.326 | 0.5784 | Kent-Corson et al. (2009) |
| Lunpola Basin | 32.06 | 89.75 | -6.763 | -6.073 | 1.920 | -0.6516 | 1.561 | -3.520 | Rowley and Currie (2006) |
| Miran River | 38.98 | 88.85 | -4.786 | -1.387 | 1.069 | -2.683 | -2.068 | 0.2831 | Kent-Corson et al. (2009) |
| Nepal Siwaliks | 27.42 | 82.84 | -1.370 | 0.006081 | -0.01583 | 0.2030 | 0.02450 | -1.588 | Quade et al. (1995) |
| Nima Basin | 31.75 | 87.50 | -5.897 | -7.724 | -0.2050 | 1.312 | 4.078 | -3.359 | DeCelles et |



| | | | | | | | | al. (2011) |
|---|---|---|---|---|---|---|---|---|
| Oiyug Basin | 29.70 | 89.50 | -10.39 | -7.842 | 2.634 | -2.598 | 1.151 | -3.735 | Currie et al. (2005) |
| Oytag | 38.98 | 75.51 | -0.4993 | -0.7157 | 1.320 | 0.7194 | -1.975 | 0.1524 | Bershaw et al. (2011) |
| Pakistan Siwaliks | 33.39 | 73.11 | 0.6450 | 0.007831 | 0.3801 | 0.4070 | 0.3792 | -0.5291 | Quade et al. (1995) |
| Puska | 37.12 | 78.60 | -2.598 | 0.006383 | 0.8959 | -0.4719 | -3.909 | 0.8806 | Kent-Corson et al. (2009) |
| Taatsin Gol | 45.42 | 101.26 | -0.7307 | -0.002622 | 1.600 | -0.3638 | -3.087 | 1.123 | Caves et al. (2014) |
| Thakkhola | 28.70 | 83.50 | -4.018 | -1.529 | 0.8024 | -0.3101 | -0.5717 | -2.409 | Garzione et al. (2000) |
| Thakkhola-Tetang | 28.66 | 83.50 | -4.018 | -1.529 | 0.8024 | -0.3101 | -0.5717 | -2.409 | Garzione et al. (2000) |
| Xiao Qaidam | 37.03 | 94.88 | 1.614 | -1.376 | 1.772 | 3.117 | -2.581 | 0.6811 | Kent-Corson et al. (2009) |
| Xifeng | 35.70 | 107.60 | 0.2447 | 0.00 | 0.5220 | 0.1725 | -0.009882 | -0.4401 | Jiang et al. (2002) |
| Xorkol | 39.01 | 91.92 | -3.218 | -0.8709 | 1.871 | -1.302 | -2.970 | 0.05405 | Kent-Corson et al. (2009) |
| Xunhua Basin | 35.90 | 102.50 | 0.4433 | -0.9613 | 1.079 | 0.3638 | -0.4095 | 0.3712 | Hough et al. (2010) |
| Yanyuan | 27.50 | 101.50 | -0.3498 | -1.152 | 0.6572 | 0.5390 | 0.3733 | -0.7672 | Hoke et al. (2014) |
| Zhada Basin | 31.50 | 79.75 | -3.983 | -4.818 | -0.04606 | 0.8307 | 2.708 | -2.657 | Saylor et al. (2009) |



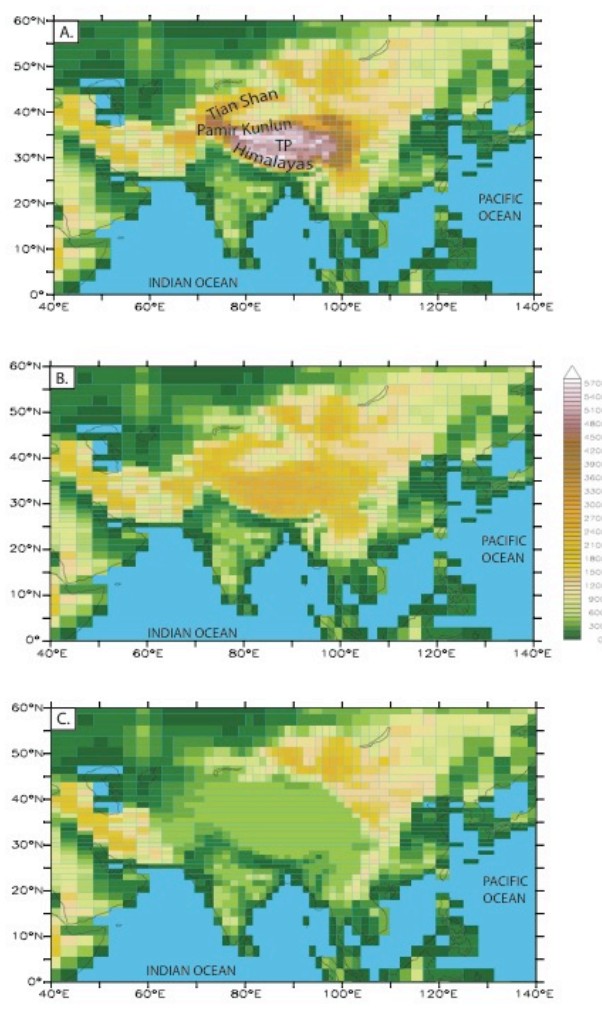

Figure 1. Models design (A) 100% of modern topography - MOD case; (B) Tibetan Plateau,
Himalayas, Tian Shan, Pamir, Kunlun and Hindu Kush elevations reduced to 50% of modern
elevation - INT case; (C) Tibetan Plateau, Himalayas, Tian Shan, Pamir, Kunlun and Hindu
Kush elevations reduced to 250 m - LOW case. Black rectangles show the division of the TP
by regions: southern TP (between 25°N and 30°N), central TP (between 30°N and 35°N) and
northern TP (between 35°N and 40°N).





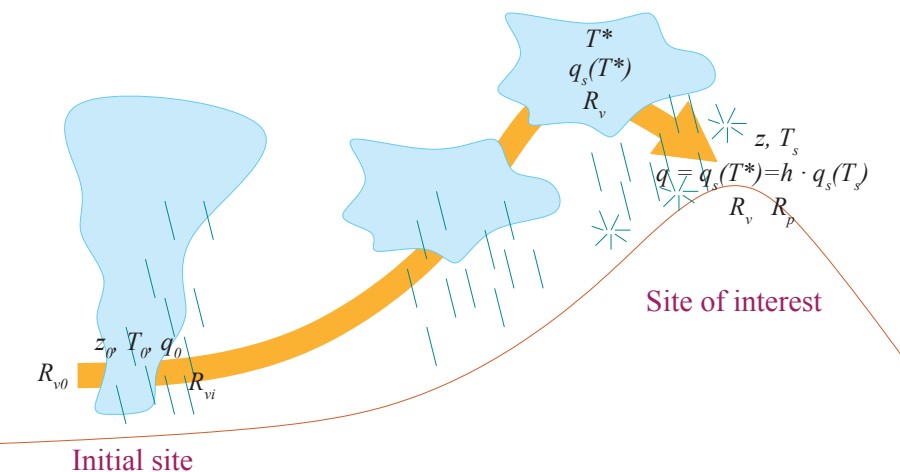

3    Figure 2. Idealized framework of an isolated air parcel transported from an initial site at low

4    altitude to the site of interest. Most notations are illustrated.





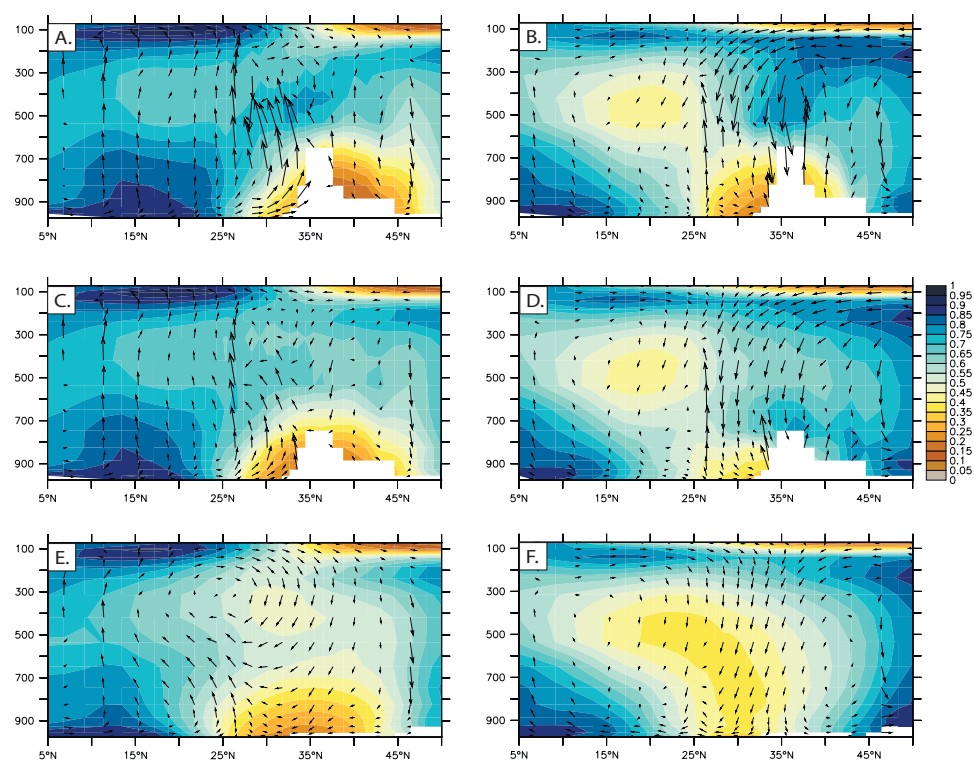

3    Figure 3. Cross-TP profiles (averaged between 70 and 90°E) showing the relative humidity

4    and moisture transport for seasons (A, C, E) MJJAS and (B, D, F) ONDJFMA and for 3

5    simulation: (A, B) MOD, (C, D) INT, (E, F) LOW cases.





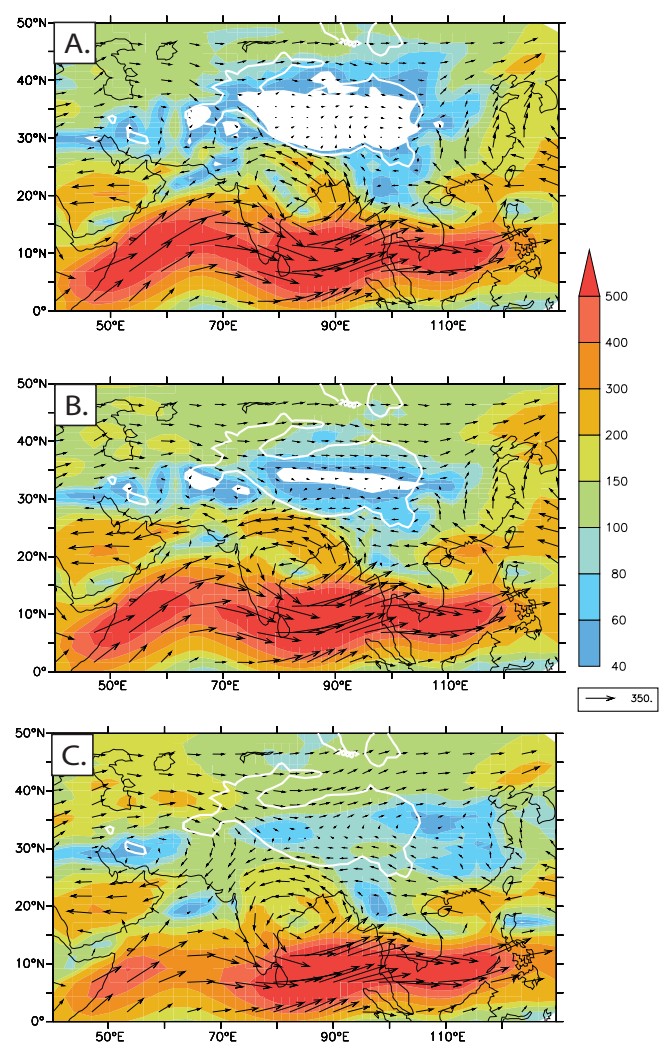

3    Figure 4. Directions and intensity of JJA vertically-integrated humidity transport for: (A)

4    MOD case, (B) INT case, (C) LOW case.





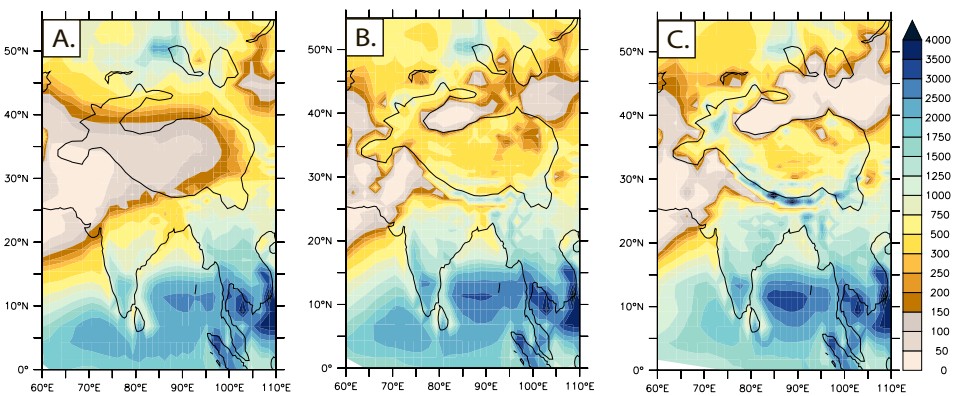

Figure 5. Annual mean precipitation amount (absolute values, mm/year) for: (A) LOW case, (B) INT case, (C) MOD case.




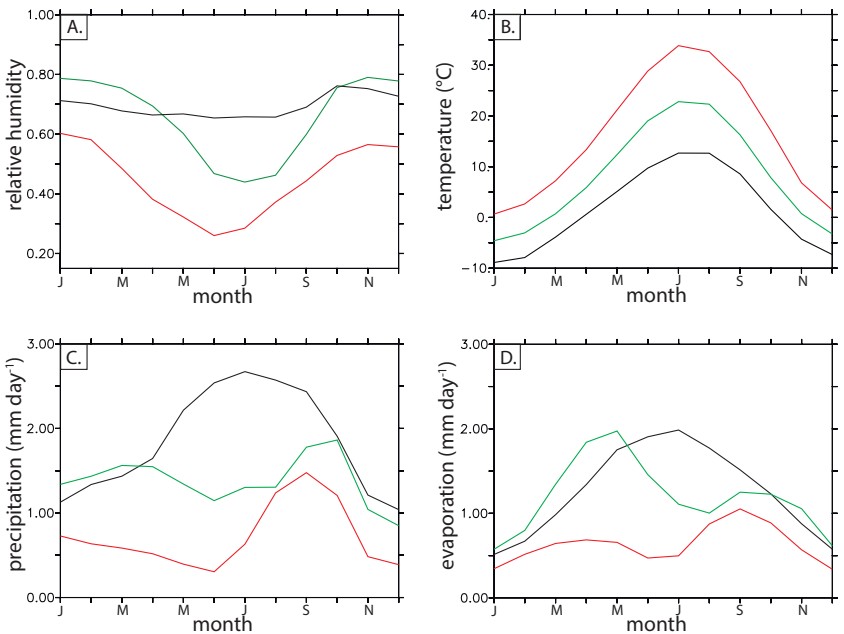

Figure 6. Intraannual variations in (A) low level relative humidity, (B) near-surface
temperature, (C) precipitation amount and (D) evaporation amount. All variables are averaged
for TP with the altitude over 1500 m. Black colour corresponds to MOD experiment, green -
for INT experiment and red - for LOW experiment.





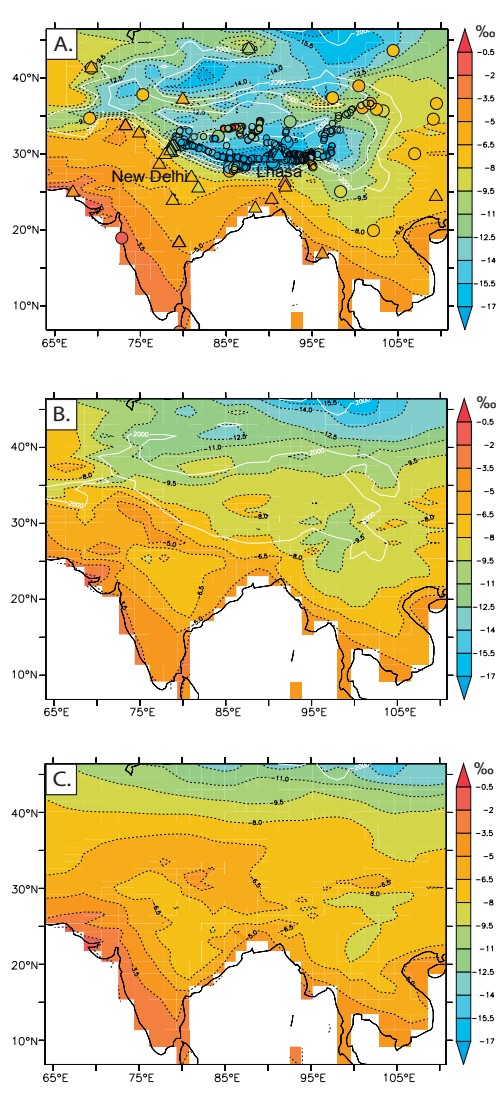

Figure 7. Annual mean $\delta^{18}$O in precipitation simulated by LMDZ-iso for: (A) MOD case, (B)
INT case, (C) LOW case. Triangles show $\delta^{18}$O in precipitation from GNIP stations, big circles
– $\delta^{18}$O in precipitation from Caves et al. (2015) compilation (annual mean and JJA values
respectively), small circles represent $\delta^{18}$O in streams, lakes and springs compiled from Quade
et al., 2011, Bershaw et al., 2012, Hren et al., 2009





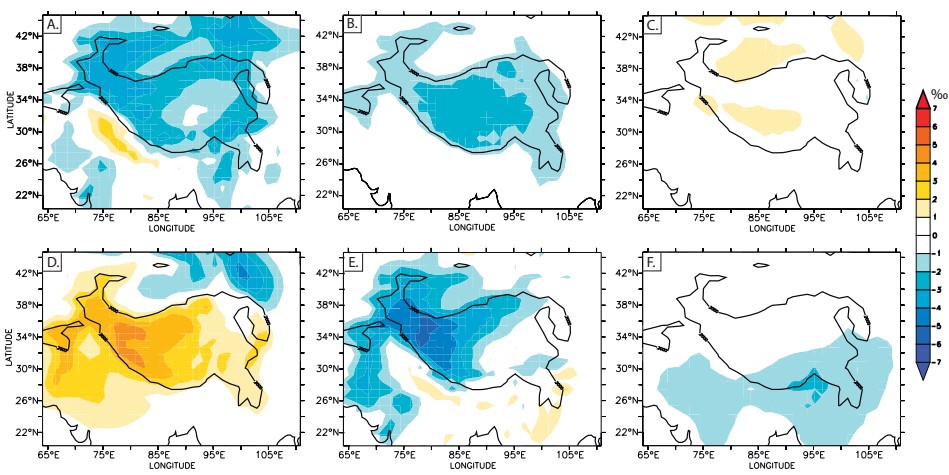

Figure 8. (A) Total isotopic difference between INT and LOW experiments ($\Delta R_p$) and spatial
isotopic variations related to: (B) direct effect of topography changes, (C) effect of lapse rate
change, associated with non-adiabatic effects, (D) effect of local relative humidity change, (E)
effect of changes in post-condensational processes, (F) all other effect (see Table 1)



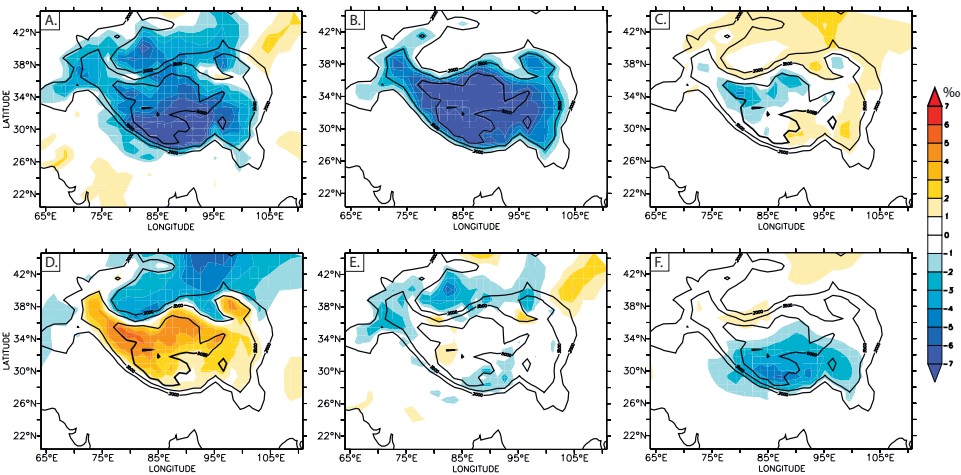

Figure 9. (A) Total isotopic difference between MOD and INT experiments ($\Delta R_p$) and spatial
isotopic variations related to: (B) direct effect of topography changes, (C) effect of lapse rate
change, associated with non-adiabatic effects, (D) effect of local relative humidity change, (E)
effect of changes in post-condensational processes, (F) all other effect (see Table 1)





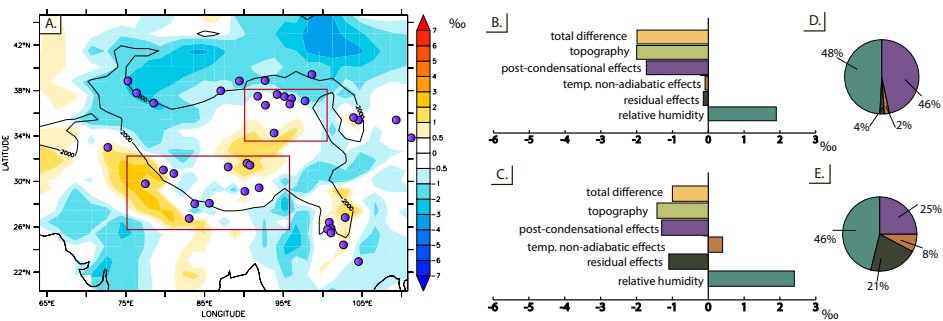

Figure 10. Difference in $\delta^{18}O_p$ between INT and LOW experiments that is not related to direct
effect of topography changes. Violet points show Cenozoic paleoelevation studies locations
(compiled from Caves et al., 2015). Red rectangles show regions for that averaged values
decomposed terms are shown: B) Northern region, C) Southern region. Pie diagrams show
portion of total isotopic difference related to processes other then topography: D) Northern
region, E) Southern region





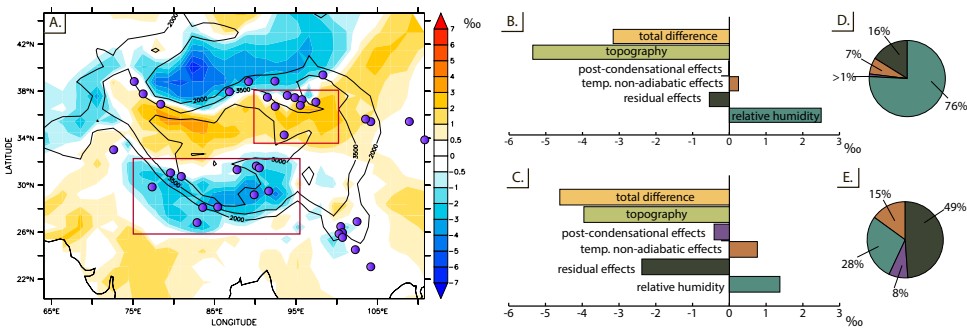

Figure 11. Difference in $\delta^{18}O_p$ between MOD and INT experiments that is not related to direct effect of topography changes. Violet points show Cenozoic paleoelevation studies locations (compiled from Caves et al., 2015). Red rectangles show regions for that averaged values decomposed terms are shown: B) Northern region, C) Southern region. Pie diagrams show portion of total isotopic difference related to processes other then topography: D) Northern region, E) Southern region.





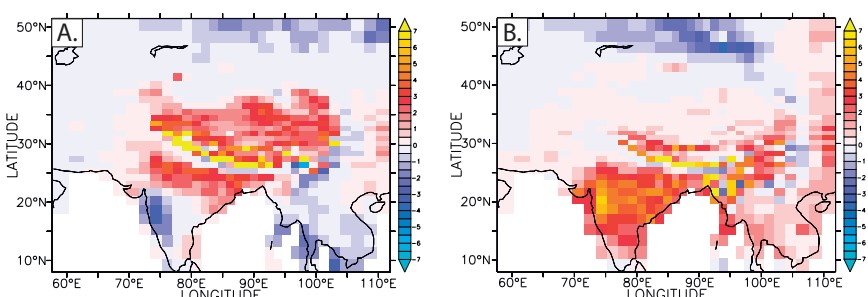

3    Figure 12. Precipitation change for A) MOD-INT B) INT-LOW cases

