# Peer review of "Impacts of Tibetan Plateau uplift on atmospheric dynamics and associated precipitation δ18O"

_Climate of the Past, 2015_

## Referee Comment (RC1) · Anonymous Referee #1 · 17 Feb 2016

The processes and factors influencing the precipitation delta 18O are very complex. This could lead to large uncertainty in paleoelevation reconstructions based on stable oxygen isotope technique. Based on model simulations, here the authors performed analyses to quantify the relative contribution of different factors related to the Tibetan Plateau uplift on the precipitation delta 18O. They also applied their results at the locations where plaeoelevation studies have been done and estimated the uncertainty in previous studies. I find this study very interesting and useful for paleoelevation reconstruction over the Tibetan Plateau region. The paper is well written and explained. I only have one question: what is the reason to choose the two topography scenarios, 50% of modern elevations and 250-m elevation? Would these specific scenarios have an influence on some of your results and conclusions? In another word, if other scenarios were chosen, would the results and conclusion be changed?

---

## Referee Comment (RC2) · Anonymous Referee #2 · 29 Feb 2016

This is an interesting and important paper. It highlights the fact the assumptions behind using del-18O as a paleo-altimetry proxy are not well satisfied and that future work should consider the importance of some of these other effects. This point has been made before for the Andes but not for Tibet. The choice of model is also very good since the variable resolution allows them to better resolve the Himalayas and Tibet.

I therefore recommend this paper for publication subject to a number of minor modifications:

(a) There is no comparison between model results and current day observations of basic climate variables (e.g. precipitation, wind etc). They need to include an extra figure on figure4 and figure 5 (and related text), and perhaps on figure 3, showing the observed humidity transport and precip for comparison. I raise this because models

often do not do an especially good job at monsoons and this may influence the confidence we have in the results in figure 8 and 9, and the numbers quoted in fig 10. However, it is unlikely to change the main message of the paper (namely that these other effects are important).

(b) Related to this, in the conclusions section, I would also like to see a brief discussion of model uncertainties. The paper concludes by advocating greater use of GCM's but does not attempt to estimate the inherent uncertainties.

(c) Is it possible to more fully explain the post-condensation processes? The causes for the changes in relative humidity were well explained but I did not understand the post condensational effects. Why did they cancel out the relative humidity changes in figure 8 (int – low) but were unimportant for figure 9 (mod-int)? The paper describes these changes but does not explain them well.

(c) On a very minor point, in some figures (e.g. fig 4) the order is mod, int, low but in other figures (e.g. fig 5) the order is reversed. Could you please keep to the same order throughout.

(d) There are no units quoted for figure 3 and 4 and 12. There is no scale for the vectors in figure 3.

---

## Author Comment (AC1) · 29 Feb 2016

We thank anonymous reviewer #1 for these comments. Indeed, we present simulations with two cases of the Tibetan Plateau elevation (a half of current topography and reduced to 250 m topography over the region) with a purpose of testing the sensitivity of $\delta^{18}$O to the topography change. Zoomed experiments including the isotopes are very expensive in terms of computation time (about 700 days of single CPU core time per experiment) and it is technically difficult to test multiple elevation scenarios. We suggest that the end-members of elevation scenarios (MOD and LOW) and the middle case (INT) are good representative of the spectrum of altitude/$\delta^{18}$O relationships. We assume that any additional scenarios with more detailed steps of relief variations would provide more information regarding altitudinal values (and possible threshold) affecting atmospheric dynamics. Still it would not change our main conclusions, i.e. that the

isotopic composition of precipitation is very sensitive to climate changes related.

Best Regards,
Svetlana Botsyun, on behalf of all co-authors
* * *

---

## Referee Comment (RC3) · Anonymous Referee #3 · 17 Mar 2016

Review of cp-2015-187

This manuscript uses an isotope-enabled GCM to understand how the many factors that influence rainfall isotopes shape the $\delta^{18}O$ of rainfall in the Himalaya-Tibet Plateau (HTP) region as topography changes. This is the first manuscript I have seen examining this topic in the HTP region although other papers have used a similar approach in S. America and elsewhere. This is an important topic to understand as it underpins the field of isotope paleoaltimetry and inferences of the topographic history of the HTP region. The paper is well written and informative, especially the author's decomposition of the isotope changes into different processes. The major comments I have relate to the mismatch between the modern and model $\delta^{18}O$ gradients on Tibet and the details of the decomposition method. I recommend acceptance after major revisions. I think the fundamental study and analysis approach is sound. However, additional explanation of the methodology and exploration of the limitations of the model are needed to make the paper useful to readers. I am also not sure their analysis will end with the same conclusions if different methods for decomposing the isotope signal are used (precip weighting climate values, non-linear adiabats that change with temperature etc.).

**Major Comments**

1. The methods for decomposing the rainfall isotope change into different components was difficult to follow. Additional details would greatly help the reader understand this process. I have a few specific notes about this below but encourage a more general re-thinking of this explanation to make it explicit how each of the terms are derived (particularly the partial differentials, reference values Rvo, To etc.).

   a. 1a. It is not directly clear how the different processes that contribute to rainfall isotope change are actually calculated. For example, how is dRvi (Eq 3) actually calculated? What region is used to define Rvi and Rvo so as to calculate dRvi? Likewise, where is q0 determined (eq 4), T0 (eq. 6) etc. A much more detailed explanation is needed I think. How is the change in elevation contribution to rainfall isotopes calculated (it depends upon dz times lapse rate but also depends upon the Rayleigh distillation, saturation relationship to T). What is the actual analytical term for the partial differential in Eq. 8?

   b. 1b. Furthermore, it is not abundantly clear how the partial differentials in eq. 8 were evaluated. For example, dRp/dRvi ($1^{st}$ partial differential in Eq. 8) depends upon f, etc. (e.g. change in rainfall Rp depends upon initial vapor composition plus modification by isotope fractionation, the magnitude in per-mil also depends upon the initial vapor composition). All of these partial differentials need to be more explicitly evaluated analytically in the paper and the dependence on the sensitivity of the partial differentials to the state at which they are calculated has to be demonstrated as not a factor (as is assumed on p7 L5).

   c. 1c. Decomposition of the absolute humidity term into temperature and relative humidity means that attribution of $\delta^{18}O$ changes to Ts or rh changes depends upon how well the model really captures those two values. Another, and more fundamental way to partition the rainfall $\delta^{18}O$ changes would be to determine how much the specific humidity changes (or T*) contribute to the isotope changes. This gets at the

transport and rainout process more directly. The local T and rh values are not really the source of the isotope changes, T* or q/qo (q[T*]/qo) are the actual cause of the changes (e.g. eq. 4). I would much rather the authors use T* or q/qo changes in their decomposition of the isotope changes (or at least do this in parallel to the T, rh decomposition as they are equivalent). In equation 8 this would combine the Dh and DdTs terms into a single q or T* term. Intuitively this makes more sense: Rp is a function of Rvi, epsilon, and q.

    d. 1d. The dz term in Eq. 8 is really a temperature change (or T* change as is assumed in a Rowley type model  - actually a Pierrehumbert-type model: Pierrehumbert, 1999 Huascaran d18O as an indicator of tropical climate during the Last Glacial Maximum).  So, the q term could be decomposed into an elevation term and a non-elevation term.  This would still keep the decomposition focused on the fundamental q/qo term rather than Ts and rh.

    e. 1e. Are the values used to decompose the rainfall isotope changes precipitation weighted?  That is, are q, T etc. weighted by the precipitation amount in a particular location?  One of the reasons that there is such a strong relationship between isotopes and elevation is that the relationship is set only when it rains.  This is a very small subset of all atmospheric conditions and thus much of the variability really doesn't matter.  It only matters if it is actually raining and the isotope signal is being "sampled."  In my opinion, rainfall weighted climate variables are essential for properly decomposing the isotope signal.

2. I think there are major and important differences in the modeled and observed rainfall isotopes (e.g. Fig. 7).  This is particularly true for the northern plateau where isotope values in rainfall become so positive that they are nearly identical to low elevation rainfall south of the Himalaya.  I disagree with the author's statement that these highly enriched isotope values are from surface processes (p9 l16-20).  Bershaw et al, 2012 were discussing the Pamir region to the west and even in their data there is not evidence of non-equilibrium fractionation as would be expected from kinetic effects.  Overall there is not a strong d-excess signal on the plateau as would be expected for evaporative processes thus the data indicate a robust positive isotope signal in rainfall values.

This feature, its interpretation, and whether and why it persists in the past are perhaps some of the major questions in Tibetan paleoaltimetry.  If it is a persistent feature then ancient isotope values from central and northern Tibet that look like today's values may have come from modern-like elevations.  If this is not a persistent feature (e.g. an arm of the Tethys north of Tibet would provide local moisture) then ancient isotope values that are the same as today may actually mean the site was at a low elevation.
My recommendations for this issue are twofold.  First, I would like to see two scatterplots in addition to the heat map.  The first would be the observed vs. modeled $\delta^{18}O$ rainfall from Fig. 7A along with RMSE estimates.  This plot would show how well the model really captures isotope values regardless of location.  The second would have latitude as an x-axis and actual observations of oxygen isotope values as the y axis along with values from the model as a continuous line.  Values could be from a swath beginning at the south and extending north along the central axis of the plateau or projected in from the plateau.  This plot would show how well the model gets the overall isotope gradient even if it doesn't get the absolute values correctly.

Second would be a thorough discussion of what this mismatch means for interpretation of the model experiments. If the model is missing or underrepresents some moisture source or process that is important today on the northern plateau then what does this mean for the conclusions from the model experiments?

3. Cite the original sources for precipitation isotope values on the plateau. These authors should get credit for the major amount of work it takes to generate this type of data and all the credit shouldn't go to the Caves 2015 compilation.

4. One of the major conclusions of the paper is that there is a non-linear effect of elevation changes on isotope values. One expects a non-linear relationship between rainfall isotope values and elevation simply because of the non-linearity of (i) saturated adiabats, (ii) the saturation vapor pressure curve with temperature and (iii) the Rayleigh distillation process itself. Thus the null hypothesis is that isotope changes with elevation from low to intermediate elevations would be less than isotope changes from intermediate to high elevations. Whether the changes are greater than can be explained by the null hypothesis needs to be demonstrated. But, the qualitative observation itself is actually expected from theory. An additional plot would drive this home. What does a Rowley (Pierrehumbert) type model predict for isotope change with elevation and where do these GCM models plot? I would focus this plot on the Himalayan mountain region as this is where the simple model is most applicable. This would be an incredibly useful plot for folks that want to take lessons away from this paper. How similar/different are the results in this paper from a simple model that has been extensively used to reconstruct elevation?

5. There is a bit of a cottage industry in the isotope-enabled GCM field looking at how isotope paleoaltimetry does/doesn't work in different orogenic systems. I would encourage the authors not to fall into the trap of saying "its complicated and you need to take additional factors into account." (This is essentially what is said at the end of the abstract.) Rather, make the information accessible and useful to the readers. Be specific and helpful in the abstract and throughout so that it is directly clear to the readers what specific factors are actually important and how they should be accounted for when reconstructing paleoaltimetry.

**Specific Comments**

P2 L16-18 – Not all of these references are carbonates or oxygen isotopes

P3 L4 – most studies do take into account changing seawater $\delta^{18}O$ either implicitly through normalization to a low elevation rainfall site or explicitly through correction using various estimates.

P3 L21 – Seems that studies of Ramstein and Fluteau should be mentioned here.

P7 L5 – Is the assumption that sensitivity of the partial derivatives to state is not important ok? It seems this would be fairly easy to test by some simple calculations and then it could be definitively stated.

P11 L1 – Are the Ddts values precipitation weighted? In general are the climatic variables used in the decomposition precipitation weighted? They should be.

P11 L7-10 – How much of the temperature changes are due to comparison to a constant adiabat for all experiments? The adiabats that matter are only ones when moisture is being transported

on to the plateau (non-linear with elevation) and the slope should change with T and qo.  Thus, inference of non-adiabatic temperature changes could simply reflect the way this is calculated and not actual changes.

P12 L28-30 – One expects a non-linear relationship between rainfall isotope values and elevation simply because of the non-linearity of (i) saturated adiabats, (ii) the saturation vapor pressure curve with temperature and (iii) the Rayleigh distillation process itself.  Thus the null hypothesis is that isotope changes with elevation from low to intermediate elevations would be less than isotope changes from intermediate to high elevations.  Whether the changes are greater than can be explained by the null hypothesis needs to be demonstrated.  But, the qualitative observation itself is actually expected from theory.

P14 L1 – Effects from post-condensation re-evaporation.  This should have a distinct d-excess signal that should be evident in the model values.  Examination of the d-excess signal spatially could directly answer this question.

P15 L1 – How do these results compare with those of Boos and Kuang (2010)?

P17 L15-16 – "Paleoelevation studies indicate the Himalayas attained their current elevation by the late Miocene."  This is not correct.  Rowley and Currie (2006) and subsequent authors indicate earlier timing for modern elevations (middle Eocene or earlier).

---

## Editor Comment (EC1) · D.-D. Rousseau (Editor) · 29 Mar 2016

Dear authors,

The third reviewer has posted an interesting review of your paper raising several key questions that I would encourage you to reply before any further move, given that my final decision to invite you for a revised version will be based on ALL your replies. So please take into consideration this third review.

All the very best

denis-didier Rousseau

CP Co-Editor in Chief

---

## Author Response (AR1)

Dear Editor,

Please find attached our revised manuscript for publication in Climate of the Past.

We addressed all reviewers points as specified in the online discussion, and provide here a stronger manuscript. More specifically, amongst the different points we addressed issues making reviewer #3 ask for major revisions :

-Validation of our model with data (this point was also raised by rev#2): We did the analyses recommended by Rev#3 and showed that the latitudinal isotopic gradient was well-represented by LMDZ-iso and that despite some model-data mismatches (that are now discussed), modeled vs. observed values gives a good fit.

- Explanation of the decomposition methods: We have rewritten the methods section to be clearer. Explanations for equations are more detailed.

- Robustness of the methods : We benefited a lot from rev#3 comments and rewrote our equations to provide a more robust decomposition. Following rev#3

suggestions, we have (1) weighted our decomposition terms by rainfall amount, (2)

checked the sensitivity to initial values, (3) checked the sensitivity to the state at which the decomposition terms have been calculated, by using a new decomposition method, based on centered differences.

Sincerely,

The authors.

**Response to anonymous review #1**

We thank anonymous reviewer #1 for these comments. Indeed, we present simulations with two cases of the Tibetan Plateau elevation (½ of current topography and reduced to 250 m topography over the region) with a purpose of testing the sensitivity of δ$^{18}$O to the topography change. Zoomed experiments including the isotopes are very expensive in terms of computation time (about 700 days of single CPU core time per experiment) and it is technically difficult to test multiple elevation scenarios. We suggest that the end-members of elevation scenarios (MOD and LOW) and the middle case (INT) are good representative of the spectrum of altitude/$\delta^{18}$O relationships. We assume that any additional scenarios with more detailed steps of relief variations would provide more information regarding altitudinal values (and possible threshold) affecting atmospheric dynamics. Still it would not change our main conclusions, i.e. that the isotopic composition of precipitation is very sensitive to climate changes related.

**Response to anonymous review #2**

We thank the Anonymous Referee #2 for this constructive review. We will provide the editor and reviewers with a corrected manuscript, meanwhile here is a point-by-point response:

(a) Model validation is obviously very important. First of all, we want to stress that LMDZ has been used for numerous present-day climate and paleoclimate studies (Kageyama et al., 2005; Ladant et al., 2014; Sepulchre et al., 2006), including studies of monsoon region (eg. (Lee et al., 2012; Licht et al., 2014). *Yao et al.,* [2013] also showed that LMDZ-iso has the best representation of the altitudinal effect compared to other GCM and RCM isotope-equipped models. These authors also have provided a detailed description of rainfall patterns over the Tibetan Plateau, and showed LMDZ-iso ability to simulate atmospheric dynamics and reproduce rainfall and $\delta^{18}$O patterns consistent with data over this region. For the purpose of our experiments validation, in the current manuscript version we compare MOD run outputs with rainfall data from the Climate Research Unit (CRU) (New et al., 2002). Corresponding figure is in the supplementary materials (Fig. S1). When compared to

CRU dataset, MOD annual rainfalls depict an overestimation over the high topography of the Himalayas and the southern edge of the Plateau, with a rainy season, which starts too early and ends too late in the year. Over central Tibet (30-35°N), the seasonal cycle is well captured by LMDz-iso, although monthly rainfall is always slightly overestimated (+0.5 mm/day). CRU data shows that the northern TP (35-40°N) is dryer with no marked rainfall season and a mean rainfall rate of 0.5 mm/day. In MOD experiment, this rate is overestimated (1.5 mm/day on annual average). In addition, we suggest to provide a comparison of humidity transport between LMDZ-iso MOD simulation outputs and ERA-40 re-analysis data (Uppala et al., 2005). This comparison depicts reasonable representation of both directions and magnitudes of moisture transport patterns by LMDZ-iso model. Model slightly overestimates the moisture transport magnitude to the west and north of the TP. Despite some model-data mismatches, the ability of LMDZ-iso to represent the seasonal cycle in the south and the rainfall latitudinal gradient over the TP as well as reasonable humidity transport allows its use for the purpose of this study. To make this comparison clearer, we will add these explanations in the corrected manuscript text and add an extra figure with model-data (with CRU precipitation) comparison to the main text and add and additions panel to the Fig. 4 with humidity transport from the ERA-40 re-analysis data.

(b) The reviewer raises a very important point. First of all, for each modelling study there are limitations associated with experimental design. In this study the topography uplift scenarios are clearly idealized, as our purpose is to test the sensitivity of $\delta^{18}O$ to the climatic changes associated with the topography uplift. For a purpose of using GCM simulations as "forward proxy modelling" (Sturm et al., 2010), realistic experiments should be designed, including accurate paleo $pCO_2$, land-sea distribution and latitudinal positions of continents. On the other hand, as it was noted by Anonymous Referee #2, $\delta^{18}O$ distribution is highly dependant of hydrological cycle representation in GCM simulations. Comparing the simulated and observed precipitation and humidity transport in the new figures, some model-data mismatches are identified. Model-data comparison show that mean annual precipitation amount is slightly overestimated by the model for the northern TP, thus could result in underestimation of the amount effect contribution for the northern TP. On the contrary precipitation model overestimates the precipitation over the southern edge of

Himalayas. If it was more realistic, the contribution of the amount effect estimated by the decomposing method would be less important. We will add a paragraph discussing these uncertainties in the conclusion section of the corrected manuscript.

(c) The difference between $\delta^{18}O_{vapour}$ and $\delta^{18}O_{precipitation}$ is linked to the post-condensation effects, mainly associated with raindrop reevaporation that can occur after initial condensation. Because lighter isotopes evaporate more easily, rain reevaporation leads to an isotopic enrichment of precipitation. Therefore, the more reevaporation, the greater the difference between $\delta^{18}O_{precipitation}$ and $\delta^{18}O_{vapour}$. We will explain this better in the paper. We refer to the study of (Lee and Fung, 2008), where post-condensation effects are explained in details. The contribution of such processes increases dramatically for very dry areas, where the relative humidity is less than 40%. In the absence of the TP (LOW experiment), large-scale subsidence superimposed to the sea surface pressure low anomaly ("Thermal Low") induces very dry condition over Asia (Fig. S2) which are favourable for high rate of post-condensational effects (Fig. 8). In contrast, the HTP uplift even to the INT height cancels the Thermal Low structure and creates relatively wet conditions (the relative humidity > 40%) over HTP with raindrop reevaporation playing a secondary role. Aridification of the Tarim Basin and creation of Taklimakan desert that is simulated for the MOD case makes post-condensational effects important over this region for the second uplift stage (Fig. 9). We agree that it is necessary to make this point clearly in the corrected manuscript.

(d) , (e) Thank you that you noticed this issue, we will change the order of figures and do necessary corrections.

Kageyama, M., Nebout, N. C., Sepulchre, P., Peyron, O., Krinner, G., Ramstein, G. and Cazet, J.-P.: The Last Glacial Maximum and Heinrich Event 1 in terms of climate and vegetation around the Alboran Sea: a preliminary model-data comparison, Comptes Rendus Geosci., 337(10-11), 983–992, doi:10.1016/j.crte.2005.04.012, 2005.

Ladant, J., Donnadieu, Y., Lefebvre, V. and Dumas, C.: The respective role of atmospheric carbon dioxide and orbital parameters on ice sheet evolution at the Eocene-Oligocene transition, Paleoceanography, 29(8), 810–823, doi:10.1002/2013PA002593, 2014.

Lee, J. and Fung, I.: "Amount effect" of water isotopes and quantitative analysis of post-condensation processes, Hydrol. Process., 22(1), 1–8, 2008.

Lee, J. E., Risi, C., Fung, I., Worden, J., Scheepmaker, R. A., Lintner, B. and Frankenberg, C.: Asian monsoon hydrometeorology from TES and SCIAMACHY water vapor isotope measurements and LMDZ simulations: Implications for speleothem climate record interpretation, J. Geophys. Res. Atmos., 117(15), 1–12, doi:10.1029/2011JD017133, 2012.

Licht, A., van Cappelle, M., Abels, H. A., Ladant, J.-B., Trabucho-Alexandre, J., France-Lanord, C., Donnadieu, Y., Vandenberghe, J., Rigaudier, T., Lécuyer, C., Terry Jr, D., Adriaens, R., Boura, A., Guo, Z., Soe, A. N., Quade, J., Dupont-Nivet, G. and Jaeger, J.-J.: Asian monsoons in a late Eocene greenhouse world, Nature, 513(7519), 501–506, doi:10.1038/nature13704, 2014.

New, M., Lister, D., Hulme, M. and Makin, I.: A high-resolution data set of surface climate over global land areas, Clim. Res., 21(1), 1–25, doi:10.3354/cr021001, 2002.

Sepulchre, P., Ramstein, G., Fluteau, F., Schuster, M., Tiercelin, J.-J. and Brunet, M.: Tectonic uplift and Eastern Africa aridification., Science, 313(5792), 1419–1423, doi:10.1126/science.1129158, 2006.

Sturm, C., Zhang, Q. and Noone, D.: An introduction to stable water isotopes in climate models: benefits of forward proxy modelling for paleoclimatology, Clim. Past, 6(1), 115–129, 2010.

Uppala, S. M., KÅllberg, P. W., Simmons, A. J., Andrae, U., Bechtold, V. D. C., Fiorino, M., Gibson, J. K., Haseler, J., Hernandez, A., Kelly, G. A., Li, X., Onogi, K., Saarinen, S., Sokka, N., Allan, R. P., Andersson, E., Arpe, K., Balmaseda, M. A., Beljaars, A. C. M., Berg, L. Van De, Bidlot, J., Bormann, N., Caires, S., Chevallier, F., Dethof, A., Dragosavac, M., Fisher, M., Fuentes, M., Hagemann, S., Hólm, E., Hoskins, B. J., Isaksen, L., Janssen, P. A. E. M., Jenne, R., Mcnally, A. P., Mahfouf, J.-F., Morcrette, J.-J., Rayner, N. A., Saunders, R. W., Simon, P., Sterl, A., Trenberth, K. E., Untch, A., Vasiljevic, D., Viterbo, P. and Woollen, J.: The ERA-40 re-analysis, Q. J. R. Meteorol. Soc., 131(612), 2961–3012, doi:10.1256/qj.04.176, 2005.

Yao, T., Masson-Delmotte, V., Gao, J., Yu, W., Yang, X., Risi, C., Sturm, C., Werner, M., Zhao, H., He, Y., Ren, W., Tian, L., Shi, C. and Hou, S.: A review of climatic controls on $\delta^{18}O$ in precipitation over the Tibetan Plateau: Observations and simulations, Rev. Geophys., 51(4), 525–548, doi:10.1002/rog.20023, 2013.

**Point by point reply to the comments from Anonymous Referee #3**

**R3:** *"This manuscript uses an isotope-enabled GCM to understand how the many factors*
*that influence rainfall isotopes shape the $\delta^{18}O$ of rainfall in the Himalaya-Tibet Plateau*
*(HTP) region as topography changes. This is the first manuscript I have seen examining*
*this topic in the HTP region although other papers have used a similar approach in S.*
*America and elsewhere. This is an important topic to understand as it underpins the field*
*of isotope paleoaltimetry and inferences of the topographic history of the HTP region.*
*The paper is well written and informative, especially the author's decomposition of the*
*isotope changes into different processes."*
A: We thank Anonymous Referee #3 for this appreciation of our work.
**R3:** *"The major comments I have relate to the mismatch between the modern and model*
$\delta^{18}O$ *gradients on Tibet and the details of the decomposition method. I recommend*
*acceptance after major revisions. I think the fundamental study and analysis approach is*
*sound. However, additional explanation of the methodology and exploration of the*
*limitations of the model are needed to make the paper useful to readers. I am also not*
*sure their analysis will end with the same conclusions if different methods for*
*decomposing the isotope signal are used (precip weighting climate values, non- linear*
*adiabats that change with temperature etc.)."*
**A:** We thank the Anonymous Referee #3 for this very constructive review. We will
provide all necessary correction in the corrected manuscript version. Meanwhile, for the
online discussion, we provide a detailed point-by-point response and a reworked methods
section in the end of this document.

**Response to Major comments:**
**R3:** *"1. The methods for decomposing the rainfall isotope change into different*
*components was difficult to follow. Additional details would greatly help the reader*
*understand this process. I have a few specific notes about this below but encourage a*
*more general re-thinking of this explanation to make it explicit how each of the terms are*
*derived (particularly the partial differentials, reference values Rvo, To etc.)."*

**A:** Thank you, we agree that the theoretical framework for the precipitation decomposition method has to be better explained. In the revised version of the manuscript we rewrite the method part with a purpose to make it clearer for readers. Full rewriting of section 2.3 is underway to make clear how each term is calculated. You will find a reworked version of sections "1. Theoretical framework for the precipitation composition" and "2. Decomposition of precipitation composition differences" below point-by-point response. We will remove the partial differentials, because it is misleading. We don't calculate partial differentials, we calculate total differences. How each term is calculated as a difference will be explicated.

**R3:** *"1a. It is not directly clear how the different processes that contribute to rainfall isotope change are actually calculated. For example, how is dRvi (Eq 3) actually calculated? What region is used to define Rvi and Rvo so as to calculate dRvi? Likewise, where is q0 determined (eq 4), T0 (eq. 6) etc. A much more detailed explanation is needed I think. How is the change in elevation contribution to rainfall isotopes calculated (it depends upon dz times lapse rate but also depends upon the Rayleigh distillation, saturation relationship to T). What is the actual analytical term for the partial differential in Eg. 8 ?"*

**A:** We add more detailed explanations and a specific equation for each analytical term from the Eq. 8 (see the end of this document). We assume that the $\delta Rvi$ term is a residual part of the vapor isotopic difference that accounts for processes of deep convection and air mass mixing. Equations from section 2 (see the decomposition method below point-by-point response) show that we do not need to know $\delta Rvi$ to estimate its contribution to the total isotopic change between two cases. For parameters z0, q0 and T0 we took values over New Delhi region to be consistent with previous isotopic studies over the region, but the sensitivity to these arbitrary choices will be tested. Even if initial conditions for the Rayleigh distillation vary depending on the atmospheric circulation, on deep convective processes and on the site of interest, we keep the same reference values, and all variations in initial conditions are accommodated by $\delta Rvi$.

**R3:** *"1b. Furthermore, it is not abundantly clear how the partial differentials in eq. 8 were evaluated. For example, dRp/dRvi ($1^{st}$ partial differential in Eq. 8) depends upon f, etc. (e.g. change in rainfall Rp depends upon initial vapor composition plus modification by isotope fractionation, the magnitude in per-mil also depends upon the initial vapor*

*composition). All of these partial differentials need to be more explicitly evaluated analytically in the paper and the dependence on the sensitivity of the partial differentials to the state at which they are calculated has to be demonstrated as not a factor (as is assumed on p7 L5)."*

**A:** To make the decomposition processes clearer we address the reviewer to the reworked methodological part (see below point-by-point response), in which we added supplemental information and equations describing the evaluation of the five partial differentials.

We carried sensitivity tests for partial differentials: results are presented in two additional tables for the southern and northern regions (Tabl. 1 and Tabl. 2). The sensitivity to Rv0 will remain as a multiplying factor. In addition, in the new version we will provide a test of the sensitivity of decomposition terms to the state at which they are calculated.

| | Northern Region | | | South region | | |
|---|---|---|---|---|---|---|
| | T0 | Rh0 | Rv0 | T0 | Rh0 | Rv0 |
| $\Delta R_{p,\Delta z}$ | 0,15 | 0,33 | 0,667 | 0,12 | 0,25 | 0,51 |
| $\Delta R_{p,\Delta\delta T_s}$ | 0,09 | 0,02 | 0,04 | 0.12 | 0.06 | 0,13 |
| $\Delta R_{p,\Delta h}$ | 0 | 0,351 | 0,66 | 0 | 0,19 | 0,83 |
| $\Delta R_{p,\Delta\delta R_{vi}}$ | 0 | 0 | 0.05 | 0 | 0 | 0,52 |
| $\Delta R_{p,\Delta\varepsilon}$ | 0 | 0 | 0 | 0 | 0 | 0 |

Tabl. 1. INT-LOW Sensitivity of the decomposition terms (in ‰) to the change of 1°C of T0 and 10 % of Rh0 and 1 ‰ of Rv0.

| | Northern Region | | | South region | | |
|---|---|---|---|---|---|---|
| | T0 | Rh0 | Rv0 | T0 | Rh0 | Rv0 |
| $\Delta R_{p,\Delta z}$ | 0,36 | 0,6 | 1,4 | 0,3 | 0,59 | 1,2 |
| $\Delta R_{p,\Delta\delta T_s}$ | 0,34 | 0,09 | 0,18 | 0,31 | 0,02 | 0,05 |

| | | | | | | |
|---|---|---|---|---|---|---|
| $\Delta R_{p,\Delta h}$ | 0 | 0,78 | 0,9 | 0 | 0,57 | 0,47 |
| $\Delta R_{p,\Delta\delta R_{vi}}$ | 0 | 0 | 0,85 | 0 | 0 | 0,67 |
| $\Delta R_{p,\Delta\varepsilon}$ | 0 | 0 | 0 | 0 | 0 | 0 |

Tabl. 2. MOD-INT Sensitivity of the decomposition terms (in ‰) to the change of 1°

of T0, 0,1 of Rh0, 1 ‰ of Rv0

**R3:** *"1c. Decomposition of the absolute humidity term into temperature and relative*

*humidity means that attribution of $\delta^{18}O$ changes to Ts or rh changes depends upon how*

*well the model really captures those two values."*

**A:** We fully agree. The question of model validation has been also raised by the

Anonymous reviewer #2. In the current manuscript version we compare MOD run outputs with rainfall data from the Climate Research Unit (CRU) [*New et al.*, 2002].

Corresponding figure is in the supplementary (Fig. S1). Also following reviewers recommendations we have added a comparison of humidity transport between LMDZ-iso

MOD simulation outputs and ERA-40 re-analysis data [*Uppala et al., 2005*]. Our MOD

simulation is preindustrial, consequently a comparison with modern data is expected to provide differences driven by the pre-industrial boundary conditions.  Still comparing

LMDZ-iso outputs with mean annual temperatures from CRU dataset [*New et al.*, 2002]

(Fig. N1) and relative humidity from NCEP-DOE Reanalysis. Fig. N2 shows that LMDZ- iso model captures these variables reasonably well.

[Figure]

Fig. N1. Mean annual temperature from A) the Climate Research Unit (CRU) [*New et al.*, 2002] dataset and B) LMDZ-iso simulated for the MOD experiment. Figure (C) represent the seasonal cycles of temperature spatially averaged from 25°N to 40°N and from 70°E to 110°E for the MOD experiment (black) and for the CRU dataset (red).

[Figure]

Fig. N2. Mean annual relative humidity profiles for A) NCEP-DOE Reanalysis and B) LMDZ-iso simulated for the MOD experiment.

**R3:** *"Another, and more fundamental way to partition the rainfall $\delta^{18}O$ changes would be to determine how much the specific humidity changes (or T\*) contribute to the isotope*

*changes. This gets at the transport and rainout process more directly. The local T and rh*

*values are not really the source of the isotope changes, T\* or q/qo (q[T\*]/qo) are the*

*actual cause of the changes (e.g. eq. 4). I would much rather the authors use T\* or q/qo*

*changes in their decomposition of the isotope changes (or at least do this in parallel to*

*the T, rh decomposition as they are equivalent). In equation 8 this would combine the Dh*

*and DdTs terms into a single q or T\* term. Intuitively this makes more sense: Rp is a*

*function of Rvi, epsilon, and q."*

**A:** One of the main purposes of the paper is to estimate the value of the $\frac{\partial R_p}{\partial z} \cdot \Delta z$ term.

Altitude acts through temperature, this is why we chose to extract the temperature signal.

We suggest to add an addition panel to the Fig. 8 and Fig. 9 that shows the part of the isotopic signal associated with the part of the change of the specific humidity with the uplift that is not associated with elevation (here Fig. N3). Nevertheless, we insist to keep all terms in the equation 8 as discussed below.

[Figure]

Fig. N3. The effect of specific humidity change

**R3:** *"1d. The dz term in Eq. 8 is really a temperature change (or T\* change as is*

*assumed in a Rowley type model - actually a Pierrehumbert-type model: Pierrehumbert,*

*1999 Huascaran d18O as an indicator of tropical climate during the Last Glacial*

*Maximum). So, the q term could be decomposed into an elevation term and a non-*

*elevation term. This would still keep the decomposition focused on the fundamental q/qo*

*term rather than Ts and rh."*

**A:** Our model is equivalent to that of Rowley et al (2001) for $\delta R_{vi} = 0$ (i.e. neglecting the effects of mixing and deep convection on the initial water vapor), $\varepsilon = (a - 1)*R_v$ (i.e.

neglecting post-condensational effects), and $h=1$ (i.e. assuming the site of interest is inside the precipitating cloud). We agree that the most important point of the paper is the division of the total isotopic signal into an "elevation" and "non-elevation" term. However, we think that the relative humidity component is important from a physical point of view because it reflects the large-scale circulation: how high did the last saturation occur? Is the regime under large-scale ascent or descent? Actually we are showing an elevation term on Fig. 8 B and Fig. 9 B according to the uplift stage and non-elevation terms of Fig. 10 A and Fig. 11 A.

**R3:** *"1e. Are the values used to decompose the rainfall isotope changes precipitation weighted? That is, are q, T etc. weighted by the precipitation amount in a particular location? One of the reasons that there is such a strong relationship between isotopes and elevation is that the relationship is set only when it rains. This is a very small subset of all atmospheric conditions and thus much of the variability really doesn't matter. It only matters if it is actually raining and the isotope signal is being "sampled." In my opinion, rainfall weighted climate variables are essential for properly decomposing the isotope signal."*

**A:** In our calculations only $\delta^{18}O$ values are weighted by the precipitation amount, but the climatic variables are not weighted. We have now recalculated all contributions using precipitation-weighted variables. In doing so, we used monthly outputs, so that the effects of seasonality are taken into account by the precipitation weighting. However, we do not have the daily outputs. So the effects of precipitation intermittency at the daily time scale won't be taken into account. We will acknowledge this limitation in the revised manuscript. Also, we checked that the decomposition terms calculated for the summer period with a large number of days when it rains is not essentially different from those calculated using mean annual values (fig. N4)

[Figure]

Fig N4. MJJAS total isotopic difference between MOD and INT experiments ($\Delta R_p$) and spatial isotopic variations related to: (B) direct effect of topography changes, (C) effect of lapse rate change, associated with non-adiabatic effects, (D) effect of specific humidity, (E) effect of local relative humidity change, (F) effect of changes in post-condensational processes, (G) all other effect, (H) difference in $\delta^{18}O_p$ between MOD and INT experiments that is not related to direct effect of topography changes

**R3:** *"2. I think there are major and important differences in the modeled and observed rainfall isotopes (e.g. Fig. 7). This is particularly true for the northern plateau where isotope values in rainfall become so positive that they are nearly identical to low elevation rainfall south of the Himalaya. I disagree with the author's statement that these highly enriched isotope values are from surface processes (p9 l16-20). Bershaw et al, 2012 were discussing the Pamir region to the west and even in their data there is not evidence of non-equilibrium fractionation as would be expected from kinetic effects. Overall there is not a strong d- excess signal on the plateau as would be expected for evaporative processes thus the data indicate a robust positive isotope signal in rainfall values."*

**A:** We agree with the reviewer that over the northern part of the Plateau there are some model-data discrepancies that could not be explained by the surface processes. On the contrary, we would like to pay attention on the very good model-data fit of isotopic data over the northern-east slope of the TP (Bershaw et al., 2011). Over the northern margins of the TP, modelled $\delta^{18}O$ in precipitation is more negative than observations show. This model-data discrepancies may result from 3 types of uncertainties: 1) linked with the model resolution. Despite quite a high resolution that we are able to obtain with a zoomed grid, the relief could be not represented well at some parts of the TP, and 2) overestimation by the model of the westerlies flux (see the comparison with the ERA moisture transport) that probably lead to underestimation of $\delta^{18}O$ over the northern part of the TP. Our statement about the contribution of the surface processes to more positive values over the central part *(p9 l16-20)* of the TP (data of Quade) is consistent with the Quade explanation of the increased role of the continental recycling northward from the Himalayas crest.

**R3:** *"This feature, its interpretation, and whether and why it persists in the past are perhaps some of the major questions in Tibetan paleoaltimetry. If it is a persistent feature then ancient isotope values from central and northern Tibet that look like today's values may have come from modern-like elevations. If this is not a persistent feature (e.g. an arm of the Tethys north of Tibet would provide local moisture) then ancient isotope values that are the same as today may actually mean the site was at a low elevation."*

**A:** We agree. In this paper we provide only sensitivity experiments with reduced topography. The influence of realistic paleogeography (eg. with the Tethys Sea, altered paleogeography) on the isotopic composition of precipitation is a topic of our further studies. However, the reviewer's point is very important and we discuss the possible sources of uncertainties while comparison with deep paleo data and further studies directions on p18 l7-10.

**R3:** *"My recommendations for this issue are twofold. First, I would like to see two scatterplots in addition to the heat map. The first would be the observed vs. modeled $\delta^{18}O$ rainfall from Fig. 7A along with RMSE estimates. This plot would show how well the model really captures isotope values regardless of location. The second would have latitude as an x-axis and actual observations of oxygen isotope values as the y axis along with values from the model as a continuous line. Values could be from a swath beginning at the south and extending north along the central axis of the plateau or projected in from the plateau. This plot would show how well the model gets the overall isotope gradient even if it doesn't get the absolute values correctly.*

*Second would be a thorough discussion of what this mismatch means for interpretation of the model experiments. If the model is missing or underrepresents some moisture source*

*or process that is important today on the northern plateau then what does this mean for*

*the conclusions from the model experiments?"*

**A:** Thank you for this recommendation. On the Fig. N5 observed vs. modeled $\delta^{18}O$

rainfall scatter plot is presented with a linear regression. Modeled vs observed data show quite a good correlation with a Person coefficient of 0,8646. Fig N6 shows a map of modeled $\delta^{18}O$ for the MOD experiment overploted by observed data values and a south- north transection (averaged between 70 and 100° E) of modelled values (black line) and projected in observed values of $\delta^{18}O$.   The general south-north isotopic gradient is simulated perfectly well by the model. After the Himalayan crest $\delta^{18}Op$ values become more positive that is consistent with a South-North trend observed by Quade et al.

[*Quade et al.*, 2007; *Bershaw et al.*, 2012b].

[Figure]

Fig. N5. Model vs observed $\delta^{18}O$ in precipitation. The colour of circles corresponds to the data set: red –

Bershaw et al, 2012, blue – Quade et al, 2011, green – Hren et al, 2009, black – Caves et al, 2015, light blue show mean annual data from GNIP stations. Red line shows a linear regression.

[Figure]

Fig. N6. A) Annual mean $\delta^{18}O$ in precipitation simulated by LMDZ-iso for the MOD case and B) S-N profiles of model simulated $\delta^{18}O$ in precipitation for the MOD case. Points correspond to present-day $\delta^{18}Op$ from published data (Bershaw et al, 2012, Quade et al, 2011, Hren et al, 2009, Caves et al, 2015), and mean annual data from GNIP stations). Solid black line shows model $\delta^{18}O$ values averaged between 70º E and 100º E. Grey lines show minimum and maximum values for the selected range of longitudes.

**R3:** *"3. Cite the original sources for precipitation isotope values on the plateau. These authors should get credit for the major amount of work it takes to generate this type of data and all the credit shouldn't go to the Caves 2015 compilation."*

**A:** Thank you for this remark. We have added the references to the original papers.

**R3:** *"One of the major conclusions of the paper is that there is a non-linear effect of elevation changes on isotope values. One expects a non-linear relationship between rainfall isotope values and elevation simply because of the non-linearity of (i) saturated adiabats, (ii) the saturation vapor pressure curve with temperature and (iii) the Rayleigh distillation process itself. Thus the null hypothesis is that isotope changes with elevation from low to intermediate elevations would be less than isotope changes from intermediate to high elevations. Whether the changes are greater than can be explained by the null hypothesis needs to be demonstrated. But, the qualitative observation itself is actually expected from theory. An additional plot would drive this home. What does a Rowley (Pierrehumbert) type model predict for isotope change with elevation and where do these GCM models plot? I would focus this plot on the Himalayan mountain region as this is*

*where the simple model is most applicable. This would be an incredibly useful plot for*

*folks that want to take lessons away from this paper. How similar/different are the results*

*in this paper from a simple model that has been extensively used to reconstruct*

*elevation?"*

**A:** We agree with the reviewer that our conclusion about non-linearity of the magnitude of the isotopic changes between the initial and the terminal stage of the uplift is a logical consequence of non-linearity of saturated adiabats, the saturation vapour pressure curve with temperature and the Rayleigh distillation process. However we find useful to stress this interesting characteristic for the geological community since such a conclusion has never been published before. The estimation whether the changes are greater than can be explained by the null hypothesis is out of scope of the paper and may be a subject of another study. Here we suggest an additional figure (Fig. N7) showing $\Delta(\delta^{18}O)$ vs elevation for MOD and INT simulations and Rowley type model (Rowley et al., 2001).

[Figure]

Fig. N7. $\Delta(\delta^{18}O)$ vs elevation for MOD (black points) and INT (blue points) and isotopic gradients for the southern region (between 25°N and 30°N). Black line shows relationship from the empirical model (Rowley, 2001; Rowley and Garzione, 2007). Green line shows second order polynomial approximation of simulated MOD $\delta^{18}O$ values. Red line shows a linear regression for the INT $\delta^{18}O$ values.

**R3 :** *" There is a bit of a cottage industry in the isotope-enabled GCM field looking at*

*how isotope paleoaltimetry does/doesn't work in different orogenic systems. I would encourage the authors not to fall into the trap of saying "its complicated and you need to take additional factors into account." (This is essentially what is said at the end of the abstract.) Rather, make the information accessible and useful to the readers. Be specific and helpful in the abstract and throughout so that it is directly clear to the readers what specific factors are actually important and how they should be accounted for when reconstructing paleoaltimetry."*

**A:** We totally agree. Text will be corrected accordingly. We also provide estimates of contribution of different decomposition terms to the total isotopic signal for locations where previous paleoelevation studies have been done in the Table. 3.

**Response to Specific Comments:**

**R3:** *"P2 L16-18 – Not all of these references are carbonates or oxygen isotopes"*

**A:** Thank you, these references has been replaced by: (Currie et al., 2005; DeCelles et al., 2007; Garzione et al., 2000; Rowley and Currie, 2006; Saylor et al., 2009; Xu et al., 2013; Li et al., 2015)

**R3:** *"P3 L4 – most studies do take into account changing seawater $\delta^{18}O$ either implicitly through normalization to a low elevation rainfall site or explicitly through correction using various estimates."*

**A:** Our purpose here was to show how climate changes have been thought to change the $\delta^{18}O$ record. We don't discuss here where these corrections used to be applied to paleoelevation reconstructions. We suggest to modify this sentence in this way: "Moreover, it has been suggested that climate-driven changes in surface ocean $\delta^{18}O$ through the Cenozoic can also influence recorded values of precipitation $\delta^{18}O$ over the continent and an appropriate corrections has been applied in most modern studies."

**R3:** *"P3 L21 – Seems that studies of Ramstein and Fluteau should be mentioned here."*

**A:** Thank you, we have added this reference.

**R3:** *"P7 L5 – Is the assumption that sensitivity of the partial derivatives to state is not important ok? It seems this would be fairly easy to test by some simple calculations and then it could be definitively stated."*

**A:** We will address this point in the revised version of the manuscript.

**R3:** *"P11 L1 – Are the Ddts values precipitation weighted? In general are the climatic*

*variables used in the decomposition precipitation weighted? They should be."*

**A:** Now we weight the decomposing terms by precipitation.

**R3:** *"P11 L7-10 – How much of the temperature changes are due to comparison to a*

*constant adiabat for all experiments? The adiabats that matter are only ones when*

*moisture is being transported on to the plateau (non-linear with elevation) and the slope*

*should change with T and qo. Thus, inference of non-adiabatic temperature changes*

*could simply reflect the way this is calculated and not actual changes."*

**A:**  We thank reviewer for making a point on this. First of all we need to note that using of one mean lapse rate (in this paper equal to $5°$ $km^{-1}$ based on the measurements of modern observed mean temperature lapse rate on the southern slope of the central

Himalayas) is clearly idealized. In our calculations we neglect the non-linearity of lapse rate. We agree that in addition to the effect of climate changes, the lapse rate is also affected by the humidity and temperature of the rising parcels. We will acknowledge this limitation in the revised version.

**R3:** *"P12 L28-30 – One expects a non-linear relationship between rainfall isotope values*

*and elevation simply because of the non-linearity of (i) saturated adiabats, (ii) the*

*saturation vapor pressure curve with temperature and (iii) the Rayleigh distillation*

*process itself. Thus the null hypothesis is that isotope changes with elevation from low to*

*intermediate elevations would be less than isotope changes from intermediate to high*

*elevations. Whether the changes are greater than can be explained by the null hypothesis*

*needs to be demonstrated. But, the qualitative observation itself is actually expected from*

*theory."*

**A:** We agree. But interestingly, up to our knowledge, this null hypothesis that isotope changes with elevation from low to intermediate elevations would be less than isotope changes from intermediate to high elevations have never been discussed before in the paleoaltimetry literature. We agree with the reviewer that this conclusion is a logic consequence of non-linearity of saturated adiabats, the saturation vapour pressure curve with temperature and the Rayleigh distillation process. However we find useful to stress this interesting characteristic for the geological community. The estimation whether the
changes are greater than can be explained by the null hypothesis is out of scope of the
paper and may be a subject of another study.

**R3:** *"P14 L1 – Effects from post-condensation re-evaporation. This should have a*
*distinct d-excess signal that should be evident in the model values. Examination of the d-*
*excess signal spatially could directly answer this question."*
**A:** Unfortunately we didn't include hydrogen isotopes in the simulations presented in this
paper. Zoomed experiments including the isotopes are very expensive in terms of
computation time (about 700 days of single CPU core time per experiment with only
oxygen isotopes) and the calculation time increases linearly with every one additional
isotope. However, with the numerical simulation we have an access to $\delta^{18}O$ in both
precipitation and vapour that gives a possibility to estimate the magnitude of post-
condensation processes without appealing to the d-excess.

**R3:** *"P15 L1 – How do these results compare with those of Boos and Kuang (2010)?"*
**A:** Although our purpose totally differs from B&K2010, our results in terms of monsoon
dynamics seem very consistent. The no-elevation run from B&K2010 depicts a weaker
monsoon and lower rainfall over Asia (Fig3b, Fig4A in their study).

**R3:** *"P17 L15-16 – "Paleoelevation studies indicate the Himalayas attained their*
*current elevation by the late Miocene." This is not correct. Rowley and Currie (2006)*
*and subsequent authors indicate earlier timing for modern elevations (middle Eocene or*
*earlier)."*
**A:** Thank you for this correction. In fact, we show that for the southern part of the TP and
Himalayas, paleoelevations based on stable oxygen isotopes measurement could be
overestimated. We will modify our text accordingly.

[revised manuscript text omitted]